# Landscape of nuclear transport receptor cargo specificity

Marie-Therese Mackmull[1] (ID), Bernd Klaus[2], Ivonne Heinze[3], Manopriya Chokkalingam[4], Andreas Beyer[4,5] (ID), Robert B Russell[6], Alessandro Ori[3,*] (ID) & Martin Beck[1,7,**] (ID)

## Abstract

Nuclear transport receptors (NTRs) recognize localization signals of cargos to facilitate their passage across the central channel of nuclear pore complexes (NPCs). About 30 different NTRs constitute different transport pathways in humans and bind to a multitude of different cargos. The exact cargo spectrum of the majority of NTRs, their specificity and even the extent to which active nucleocytoplasmic transport contributes to protein localization remains understudied because of the transient nature of these interactions and the wide dynamic range of cargo concentrations. To systematically map cargo–NTR relationships *in situ*, we used proximity ligation coupled to mass spectrometry (BioID). We systematically fused the engineered biotin ligase BirA* to 16 NTRs. We estimate that a considerable fraction of the human proteome is subject to active nuclear transport. We quantified the specificity and redundancy in NTR interactions and identified transport pathways for cargos. We extended the BioID method by the direct identification of biotinylation sites. This approach enabled us to identify interaction interfaces and to discriminate direct versus piggyback transport mechanisms. Data are available via ProteomeXchange with identifier PXD007976.

**Keywords** interaction network; nuclear pore complex; protein transport; proteomics; proximity ligation

**Subject Categories** Genome-Scale & Integrative Biology; Network Biology; Post-translational Modifications, Proteolysis & Proteomics

**Mol Syst Biol. (2017) 13: 962**

## Introduction

The nuclear transport system can be roughly grouped into a stationary and a soluble phase. Nuclear pore complexes are stationary and fuse the inner and outer nuclear membranes to form aqueous channels across the nuclear envelope (reviewed in Beck & Hurt, 2017). They are made up of ~30 nucleoporins (Nups) that can be further subdivided into two categories. Firstly, the scaffold Nups of the Nup107 and Nup93 subcomplexes are comprised of folded domains and form the architectural elements cylindrically grouped around the central channel of the NPC (Fig 1A). Secondly, FG-Nups are anchored to scaffold Nups and line the central channel. They contain intrinsically disordered stretches rich in phenylalanine–glycine (FG) repeats that form the permeability barrier that selectively excludes non-inert macromolecules from the central channel, while small molecules and proteins can passively diffuse across the NPC. The FG-rich domains of the Nup214 and Nup358 complexes together with Nlp1 are anchored to the cytoplasmic ring. The respective domains of Nup153 and Nup50 comprise their counterpart at the nuclear ring. In contrast, the FG domains of the Nup62 complex and possibly also Nup98 and POM121 are symmetrically anchored into the inner ring (Fig 1A). Although the exact role of the different types of FG-Nups has to be further investigated, it has been well established that FG-Nups interact with the soluble phase of the nuclear transport system thereby enabling active nucleocytoplasmic exchange (Fig 1B; Radu *et al*, 1995; Strawn *et al*, 2004). This soluble phase consists of the small GTPase RAN, a number of auxiliary factors as well as nuclear transport receptors (NTRs) that transiently interact with FG-repeats but also explore the open space of cytoplasm and nucleoplasm to recruit cargos. The human genome harbors 20 NTRs of the importin β family (Kimura & Imamoto, 2014) that are HEAT repeat-containing proteins. Most of them recognize different types of nuclear localization signals (NLSs) or nuclear export signals (NESs) of cargo proteins and facilitate their import into or export from the nucleus, respectively (Fig 1B).

Depending on the directionality of cargo transport, NTRs are referred to as importins and exportins, although this strict subdivision does not hold true for all of them. For example, importin 13 (IPO13) has also been shown to have export capacity for the translation initiation factor elF1A (Mingot *et al*, 2001), and also exportin 4 (XPO4) has a bidirectional transport capability (Gontan *et al*, 2009).

1   Structural and Computational Biology Unit, European Molecular Biology Laboratory, Heidelberg, Germany
2   Centre for Statistical Data Analysis, European Molecular Biology Laboratory, Heidelberg, Germany
3   Leibniz Institute on Aging, Fritz Lipmann Institute (FLI), Jena, Germany
4   Cellular Networks and Systems Biology, CECAD, University of Cologne, Cologne, Germany
5   Center for Molecular Medicine Cologne, University of Cologne, Cologne, Germany
6   Heidelberg University Biochemistry Centre & Bioquant, Heidelberg, Germany
7   Cell Biology and Biophysics Unit, European Molecular Biology Laboratory, Heidelberg, Germany
    *Corresponding author. Tel: +49 3641 656808; E-mail: alessandro.ori@leibniz-fli.de
    **Corresponding author. Tel: +49 6221 3878367; E-mail: martin.beck@embl.de

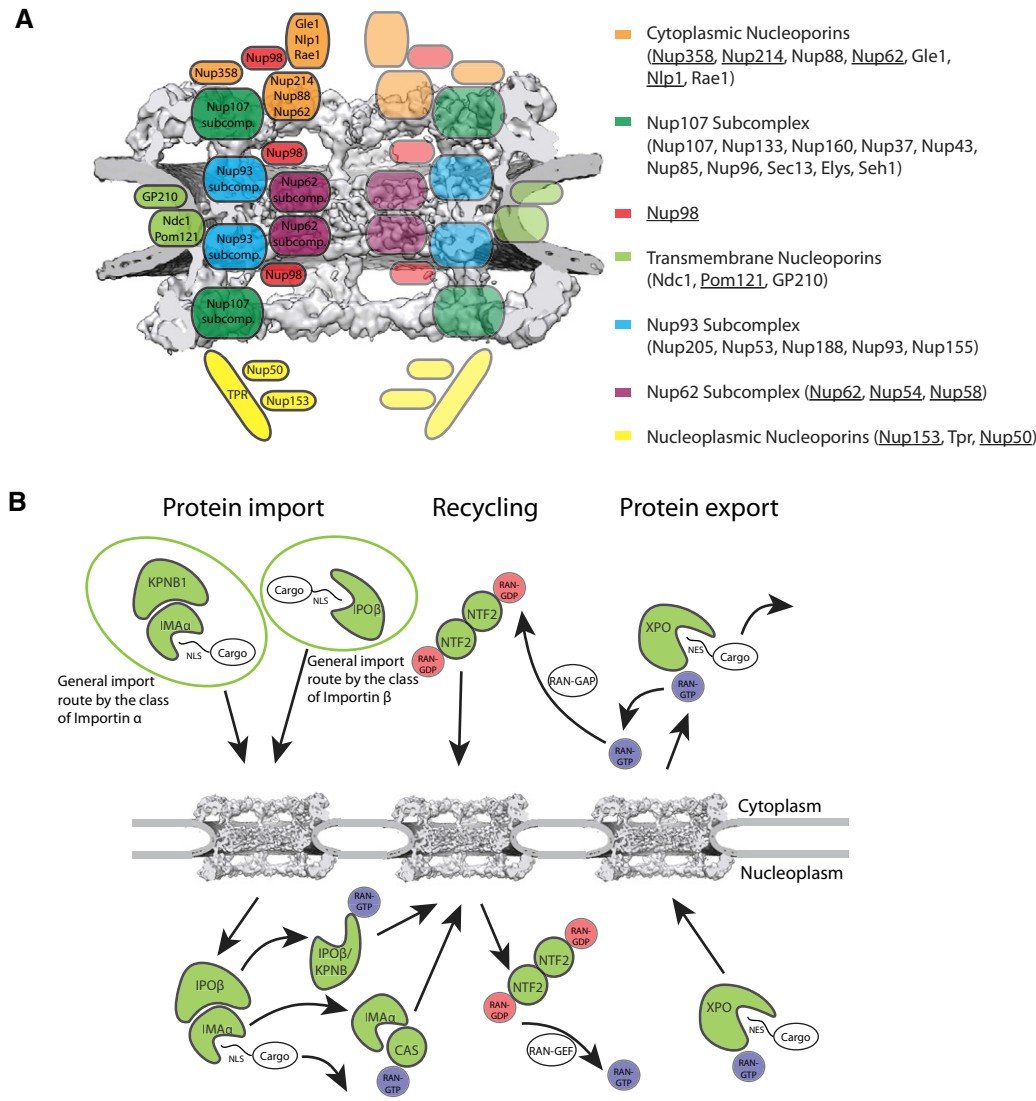

**Figure 1. Scheme of the nucleocytoplasmic transport system.**

A   Scheme showing the composition and approximate position of nucleoporin subcomplexes with respect to the overall structure (Protein Data Bank EMD-3103). FG-Nups are underlined.

B   Overview of the nucleocytoplasmic transport pathways and NTRs. Cargos are either bound by importin βs and exportins or bound by importin αs that serve as an adaptor for KPNB1. Recycling of RAN-GDP by NTF2 into the nucleus and importin αs by XPO2 back to the cytoplasm is shown.

A specialized member of the importin β family is the cellular apoptosis susceptibility (CAS/XPO2) protein that is a recycling factor for importin αs (Kutay *et al*, 1997). NTRs of the importin α family consist of 10 armadillo (ARM) repeats and an importin β binding (IBB) domain. They often, but not always (Kotera *et al*, 2005), function as adaptor proteins, and form a ternary import complex together with their cargo and importin β (Görlich *et al*, 1995). Three clades of importins of the α-type occur in vertebrates, and humans possess at least seven distinct importin αs. Although they are differentially expressed across tissues and during development (Yasuhara *et al*, 2009), they all bind classical NLSs (cNLSs) and are thought to be to some extent functionally redundant (Goldfarb *et al*, 2004; Yasuhara *et al*, 2009; Kelley *et al*, 2010). The small GTPase RAN fuels the active nuclear transport cycle. The binding to RAN-GTP triggers the disassembly of import complexes at the nuclear side of the NPC (Izaurralde *et al*, 1997), and the formation of export complexes either with cargo or importin α or β for recycling (Fig 1B). The disassembly of each of these export complexes is triggered by GTP hydrolysis in the cytoplasm. RAN-independent transport pathways also have been reported (Miyamoto *et al*, 2002; Kotera *et al*, 2005).

The nuclear transport system controls the nucleocytoplasmic localization of essential cellular components by actively transporting them across the nuclear envelope. It facilitates the import of transcription factors that function downstream of signaling pathways, the import of histones that are required for DNA replication, and the export of ribosomes and fully processed mRNAs needed for translation and, ultimately, cell proliferation (Dickmanns *et al*, 2015).

Therefore, the nuclear transport system critically contributes to regulating cellular homeostasis. It is also involved in key events during development (Yasuhara *et al*, 2007; Okada *et al*, 2008) and malignant transformation (Winkler *et al*, 2016). Indeed various clinically manifested mutations map into components of the nuclear transport system, and alterations of transport pathways have been linked to various human diseases (reviewed in Kimura & Imamoto, 2014).

To understand how the nuclear transport system contributes to protein localization, knowledge of its cargo spectrum is absolutely essential. However, several technical obstacles have rendered the systematic mapping of cargo–NTR relationships challenging: (i) The transient nature of cargo–NTR interactions makes them largely inaccessible to biochemical identification by NTR-centric affinity purification; (ii) each NTR does recognize a multitude of cargos and their cargo spectrum might overlap; and (iii) cargo abundances (copy numbers per cell) span several orders of magnitude *in situ*. The vast majority of previous studies therefore were designed in a cargo-centric manner and have reported transport pathways for individual cargos (for review, see, e.g., Chook & Süel, 2011; Kimura & Imamoto, 2014; Twyffels *et al*, 2014). Although powerful for studying the contribution of the nuclear transport system to individual cellular mechanisms, these approaches are not suitable to comprehensively chart its cargo spectrum. Systematic approaches to measure the cargo spectrum in an NTR-centric manner have been reported, but were often limited to a single NTR (XPO1; Thakar *et al*, 2013; Kırlı *et al*, 2015), or conducted *ex vivo*. The import rate of SILAC-labeled cellular extracts has been measured using mass spectrometry in the presence and absence of various importin βs to chart their cargo spectrum (Kimura *et al*, 2013a, 2014, 2017). This work was performed in permeabilized cells in which the cytoplasmic membrane has been punctured and the cytosolic content washed out. Cellular extracts were biochemically depleted for the soluble phase of the nuclear transport system, and selected recombinant transport factors were added back to study their cargo spectrum. In case of XPO1, its susceptibility to inhibitors (Nishi *et al*, 1994) and dependency on RAN-GTP for export complex formation have been exploited to systematically measure its cargo spectrum (Kırlı *et al*, 2015), but these approaches are not generalizable to all NTRs. The cargo spectrum of the vast majority of NTRs remains incompletely charted. It remains understudied to which extent the functions of the different importins, exportins, and transportins are distinct or functionally redundant and to which extent the nuclear transport system contributes to protein localization. Experiments with the XPO1 inhibitor leptomycin B suggested that only a small fraction of the nuclear proteome is actively exported in *Xenopus* oocytes (Wühr *et al*, 2015), while aforementioned biochemical study has identified a rather large number of XPO1 cargos (Kırlı *et al*, 2015). Whether these results are generically applicable to somatic cells or specific to the very special cell properties of oocytes remains to be further explored.

Here, we have systematically measured the *in situ* interactome of NTRs using proximity ligation mass spectrometry based on the BioID system (Roux *et al*, 2012). We fused about half of all human NTRs to BirA*, an engineered biotin ligase that covalently modifies even transient interactors that were subsequently affinity purified and identified by mass spectrometry (Table EV1). We implemented novel approaches to estimate the specificity of interactions and false discovery rate (FDR) in BioID experiments. We exploited the direct identification of biotinylated peptides to reveal previously unappreciated interactions between Nups and NTRs. We validated our data experimentally and against previously published, independent data sets, and report a multitude of NTR interactions. We demonstrate that the different subunits of protein complexes are often proximate to the same NTRs, stressing that large complexes are most often transported as a whole. Our data point to a scenario in which a considerable fraction of the human proteome is subject to active nuclear transport. We provide a comprehensive resource that shall be invaluable to various scientific communities.

# Results

## Proximity ligation is a powerful method to chart transient interactions of nuclear transport receptors *in situ*

To systematically chart the NTR interactome, we targeted various human NTRs using the BioID system. The selected NTRs broadly cover the different types of active nuclear transport and include three importin αs (IMA1, IMA5, and IMA6) together with the respective adaptor importin β1 (KPNB1), 4 other importin βs (IPO4, IPO5, IPO11, and IPO13), two transportins (TNPO1 and TNPO2), three exportins (XPO1 also called CRM1, XPO7, and XPO2 also called CAS or CSE1L) as well as a few auxiliary transport factors like RAN, the RAN recycling factor NTF2, NXT1, and NXT2 (Table EV1). Due to the proteomic nature of our methodology, we decided to primarily focus on protein transport pathways and to neglect RNA export, for which the substrates are often obvious, for example, because mRNAs are translated in the cytoplasm, but non-straight forward to identify by mass spectrometry.

We tagged NTRs with the engineered biotin ligase BirA* and inducibly expressed the fusion proteins in stable human embryonic kidney (HEK293) cell lines for 24 h (Fig 2A). In agreement with previous work (Roux *et al*, 2012), we found that after additional 24 h of exposure to biotin, the labeled proteins had accumulated. We monitored the subcellular localization of the tagged NTRs and biotinylated proteins using fluorescence microscopy based on anti-FLAG- and streptavidin-staining, and found that the different BirA* fusion proteins label specific subcellular compartments (Fig 2B). For example, importins primarily labeled nuclear proteins and, to a lesser extent, the nucleolus. XPO7, NTF2, and a control fusion protein that shuttles between cytoplasm and nucleoplasm (NLS-NES-Dendra2, see below) labeled both, cytoplasmic and nucleoplasmic components. Non-engineered cells displayed only a faint staining correlating with mitochondria that contain a few naturally biotinylated enzymes that are involved, for example, in fatty acid synthesis (Waite & Wakil, 1966). We conclude that the subcellular localization of the labeled proteins correlates with known properties of the respective NTRs.

We collected four biological replicates (independent isolations from cells at different passages) from each of the NTR fusions and four control cell lines, namely BirA* alone, the aforementioned shuttling NLS-NES-Dendra2 with C- or N-terminal BirA* tag, and non-engineered cells (Fig EV1A). Biotinylated proteins were affinity purified using streptavidin beads (Fig 2A). Because of the very high affinity of biotin for streptavidin, previous studies relied on on-bead

**Figure 2. Proximity ligation of nuclear transport receptors.**

A Scheme of the general workflow used in this study showing the molecular cloning, proximity ligation *in situ* and affinity capture of biotinylated proteins on streptavidin sepharose beads. Isolated proteins were on-bead digested using trypsin for indirect identification and biotinylated peptides eluted in an additional step using a mixture of ACN and TFA for direct identification, both by MS.

B Stable cell lines expressing transport factors fused to BirA* were fixed and stained with Streptavidin-Alexa 647 (red) to visualize the subcellular localization of biotinylated proteins, the overexpressed BirA* fusion proteins (anti-FLAG, green), and the DNA (Hoechst, blue). Scale bar, 10 μm.

digestion of the captured proteins. Thereby, primarily non-biotinylated peptides elute after tryptic digestion and the labeled proteins are identified indirectly. Here, we introduced a subsequent elution step under harsh conditions using organic solvents (see Materials and Methods and Ori *et al*, 2009) to effectively elute the biotinylated peptides that are then identified directly. The combined approach is not only very effective to identify even transient interactors of NTRs, but it also provides the exact biotinylation sites as an additional layer of information.

We measured tryptic peptides from on-bead digestion by tandem mass spectrometry in technical duplicates and used label-free quantification to compare the spectrum of the interacting proteins for each NTR. Even low abundant interactors were detectable, and the whole workflow was highly reproducible (average Pearson correlation coefficient between biological replicates $R > 0.96$, Appendix Fig S1A and B). In order to assess the impact of BirA* fusion on NTR interactions, we generated two independent cell lines for most NTRs by fusing BirA* either N- or C-terminally, whenever possible. We observed that in the majority of cases, the nature of the BirA* fusion did not influence the interaction spectrum of the NTRs. In such

cases, N- and C-terminal fusions of the same NTR displayed correlation coefficients that were marginally lower than biological replicates obtained from the same fusion construct (Appendix Fig S1C). In some cases, though, the localization of BirA* had an impact on the abundance of the identified proteins suggesting a perturbation of NTR function. For example, we observed a comparably lower correlation between the interactions of BirA*-IMA1 and IMA1-BirA* ($R = 0.82$, Appendix Fig S1C), possibly because binding to importin β was sterically hindered (see Discussion). Taken together, the BioID system provides powerful means to monitor transient interactions of components of the nucleocytoplasmic transport system.

**Draft of the actively transported proteome**

From a total number of 32 isolations (total of 256 MS runs including control samples and technical duplicates, Fig EV1A), we identified ~4,000 protein groups using stringent criteria (global protein group FDR < 1% and at least two unique peptides identified in at least three biological replicates per cell line), which comprises more than

one-third of the proteome typically expressed in human tissue culture cells (Geiger *et al*, 2012). Of course, this set includes contaminant identifications such as proteins that non-specifically bind to the streptavidin beads but also other interactors such as for example components of the nucleocytoplasmic transport system that interact with NTRs without being necessarily cargos.

To deduct the subset of proteins involved in active nucleocytoplasmic transport, we compared each NTR-BirA* fusion against the control set in a pairwise fashion. The control set was generated by combining replicates from four samples (BirA* alone, BirA*-NLS-NES-Dendra2, NLS-NES-Dendra2-BirA*, and non-engineered cells; 16 independent experiments in total, Fig EV1B). These control samples were selected to take into account naturally biotinylated proteins (non-engineered cells), unspecific biotinylation of potential BirA*-interacting proteins (BirA* alone control), and background arising from the nucleocytoplasmic transport system (NLS-NES-Dendra2 controls). All pairwise comparisons were combined and false-positive identification rates globally calculated for the entire set (using Sime's method, see Materials and Methods for details). About 33% (1,252) of all identified proteins were found to be significantly enriched in at least one NTR fusion sample and are further considered as the NTR-interacting proteome (NIP), while about 17% (623) were enriched in the control samples, considered as the background proteome (adjusted *P*-values < 0.1 after Benjamini and Hochberg correction for multiple testing; Table EV2).

Gene Ontology (GO) analysis revealed the expected pattern of cellular compartment enrichment for the NIP and background proteome: The NIP is strongly enriched for nuclear proteins while background proteome tend to be of mitochondria/cytoplasmic origin (Fig 3A, Table EV3). The enrichment for mitochondria in background proteome is likely caused by the occurrence of naturally biotinylated proteins localized in this organelle. In case of the GO category "Molecular Function", the NIP was strongly enriched for functions annotated as DNA-, RAN-GTPase-, or zinc ion-binding (Fig 3B, Table EV3). Probably related to the latter, transcription factor activity and regulatory activities were also enriched in the NIP. On the other hand, GO terms enriched in the background proteome were mainly associated with functions typical of mitochondrial proteins (Fig 3B). We next compared our results with previous data of nucleocytoplasmic partitioning (Wühr *et al*, 2015), where each protein was assigned to the RNC (relative nuclear concentration) calculated for over 9,000 proteins of *Xenopus laevis* (Fig 3C). We found that the NIP shows a significant enrichment of proteins with high RNC values, indicative of asymmetric distribution with higher nuclear concentration. The background proteome showed an opposite trend with RNC scores indicating a predominant cytoplasmic localization. Proteins with a molecular weight (MW) above 30–50 kDa need a NTR to be actively transported through the NPC. We compared MW distributions between NIP and background proteome proteins (Fig 3D). NIP proteins indeed display a MW distribution shifted toward higher MW in comparison to the background proteome. Taken together our data provide evidence of specific interactions with NTRs for over 1,200 proteins that comprise one-third of all identified proteins. This set includes various known actively transported proteins, but also a large fraction of proteins for which we provide the first experimental evidence of NTR interactions. Since our study is restricted to a specific cell type and biological condition, we believe that a considerably higher number of proteins might be subject to active nuclear transport.

## Functional redundancy and specificity of NTRs

The above-described analysis globally identifies the actively transported proteome in a conservative manner but it falls short in assessing the functional redundancy of the different NTRs. To address this aspect, we took advantage of our large data set and calculated interaction specificity scores for each protein by comparing NTR samples against each other (Fig EV1C, see Materials and Methods for details). The specificity score is derived by combining *P*-values from all the possible pairwise comparisons for a given NTR using the Fisher method. Subsequently, a global FDR correction was applied based on all tests performed. This allowed us to define for each NTR a subset of proteins that show statistically significant enrichment as compared to other NTRs. This specific interactome of each NTR contains candidate substrates for active nuclear transport that we will subsequently refer to as cargos. Candidates that were validated by an independent method we will refer to as validated cargos.

We typically identified between 400 (BirA*-TNPO1) and 800 (IMA1-BirA*) significantly enriched proteins per NTR (adj. Fisher *P*-value < 0.01; Appendix Fig S1D, Table EV4). 30% (~1,100) of the enriched proteins are significant to maximum two NTR samples, often the N- and C-terminally fused version of the same NTR, and 60% (~2,200) are significant in maximum four NTR samples (Appendix Fig S1F). The global correlation of specificity scores shows distinct clusters of functionally related NTRs (Fig 4A) and further verifies the consistency of data derived from N- and C-terminal fusion proteins (Fig 4A and B). We quantified the degree of functional redundancy between different NTRs in combined C- and N-terminal data sets (Appendix Fig S2A). A high degree of redundancy is found between importin αs, between IPO4 and IPO5 (Appendix Fig S2A). Specifically, IMA1-BirA* and IMA5-BirA* share 478 of their interacting proteins, 179 of which are also detected with IMA6-BirA* (Fig 4C). Importin αs depend on their interaction with importin β (KPNB1) for the transport of cargoes across the NPC. Our data recapitulate this functional interaction in two ways: (i) Multiple importin αs are found among the specific interactors of KPNB1, and (ii) more than 50% of the interactions of BirA*-KPNB1 are shared with importin αs and thus likely to be mediated by these adaptor proteins (Fig 4E and D). The three exportins investigated here show little overlap among themselves (Appendix Fig S2A). Instead, the broadly acting XPO1 shows a certain degree of overlap with all investigated transport factors, while XPO2 shows redundancy with importin αs, for which it is the recycling factor (see Discussion). Interestingly, the specific interactome of XPO7 overlaps with IPO4 and IPO5, suggesting a possible functional relationship.

## Validation of cargos and localization signals

In line with previous large-scale studies that validated their cargos using independent methods (Kimura *et al*, 2017), we wanted to assess the recall of known NTR cargos by comparing protein abundance levels in the BioID experiments versus specificity scores. Bait proteins are generally retrieved as highly abundant and specific targets owing to the ability of the BirA* fusion proteins to

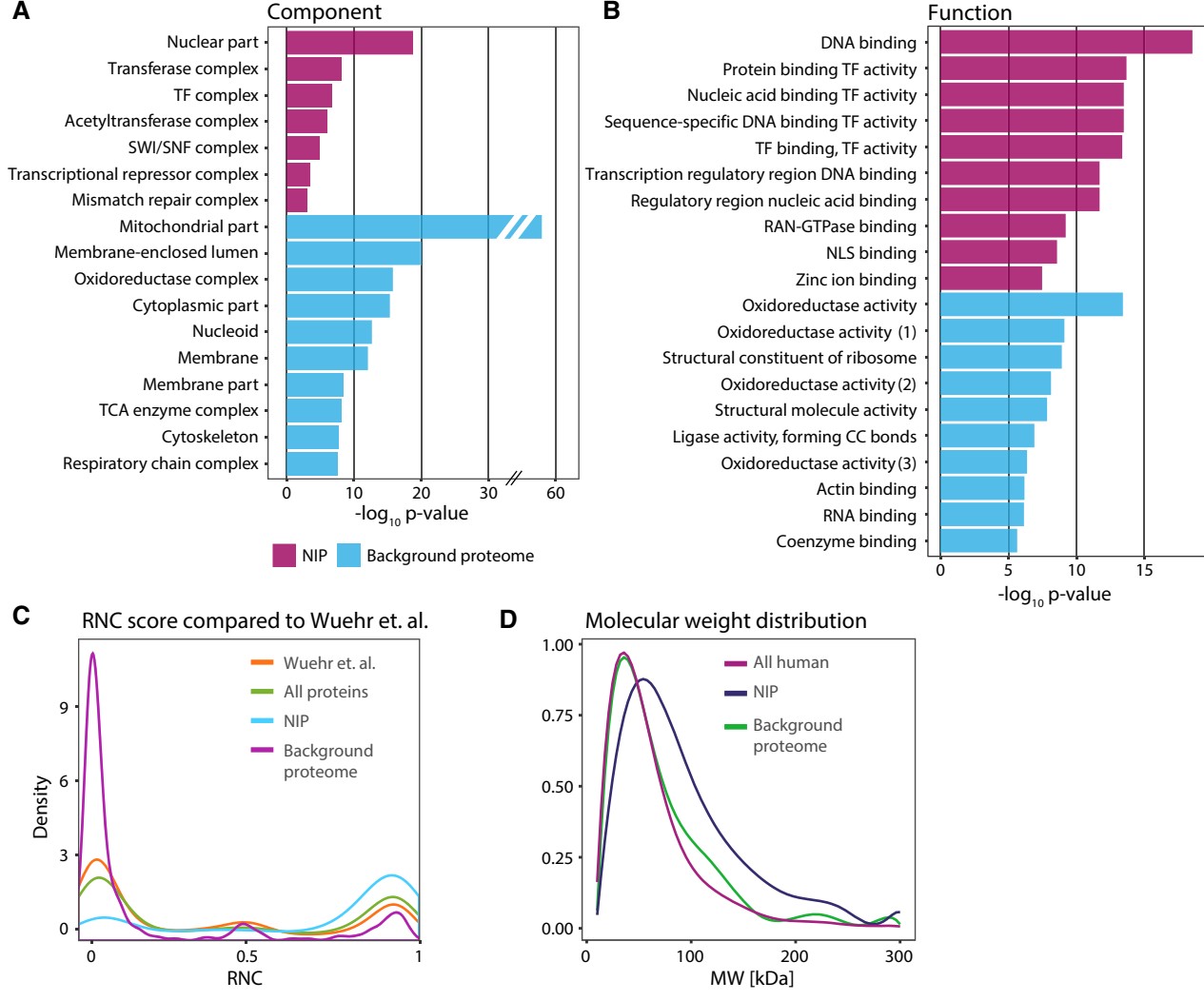

**Figure 3.  Subcellular localization and function of the NTR-interacting proteome (NIP) and background proteome.**

A, B   GO enrichment analysis for the terms "cellular component" (A) and "molecular function" (B). The top 10 most significant GO terms are plotted for the NIP or background proteome. Numbers in (B) refer to (1) the aldehyde or oxo group of donors, (2) the aldehyde or oxo group of donors, NAD or NADP as acceptor, (3) NAD(P)H. The NIP is significantly enriched for nuclear proteins.

C   RNC score from Wühr *et al* (cytoplasmic to nuclear from 0 to 1) compared to all significantly identified proteins in the NIP and background proteome. The NIP has a trend toward high RNC values indicating a high content of proteins that show preferential nuclear localization, while to opposite is the case for the background proteome.

D   MW distribution of all proteins in the NIP (median = 74 kDa) and background (median = 54 kDa) compared to all the reviewed human proteins in the UniProt database (median = 46 kDa). The NIP shows a trend toward higher MW even though protein complex association (so-called native MW) is not considered in this plot.

biotinylate themselves (Fig 4E and F, and Appendix Fig S2B). Specific cargos, such as the eIF3 complex in the case of BirA*-IPO5, display high specificity score and high abundance. On the other hand, common components of the nucleocytoplasmic transport system such as nucleoporins, which interact with all the NTRs, display medium/high abundance, but low specificity scores (Fig 4F).

Our data recapitulate many known interactions of NTRs with other components of the nuclear cytoplasmic transport system. These include the interaction between XPO1 and snurportin-1 (Paraskeva *et al*, 1999), the interaction between NXF1 and NXTs

(Wiegand *et al*, 2002), and the reciprocal specificity between XPO2 and multiple of importin α (Fig 4G and Appendix Fig S2B). We also compared cargos identified in large-scale studies for several importin βs (Fig 4H and Appendix Fig S2C; Kimura *et al*, 2017) and XPO1 (Fig 4I; Thakar *et al*, 2013; Kırlı *et al*, 2015; Wühr *et al*, 2015). The three large-scale studies performed for XPO1 had limited overlap with each other in terms of identified cargos (maximum 29%). We therefore decided to compare our data to validated cargos that were identified in at least two out of three studies (71 proteins). 38% of these cargoes were also identified by our approach to be specifically interacting with XPO1 (Fig 4I). These include validated

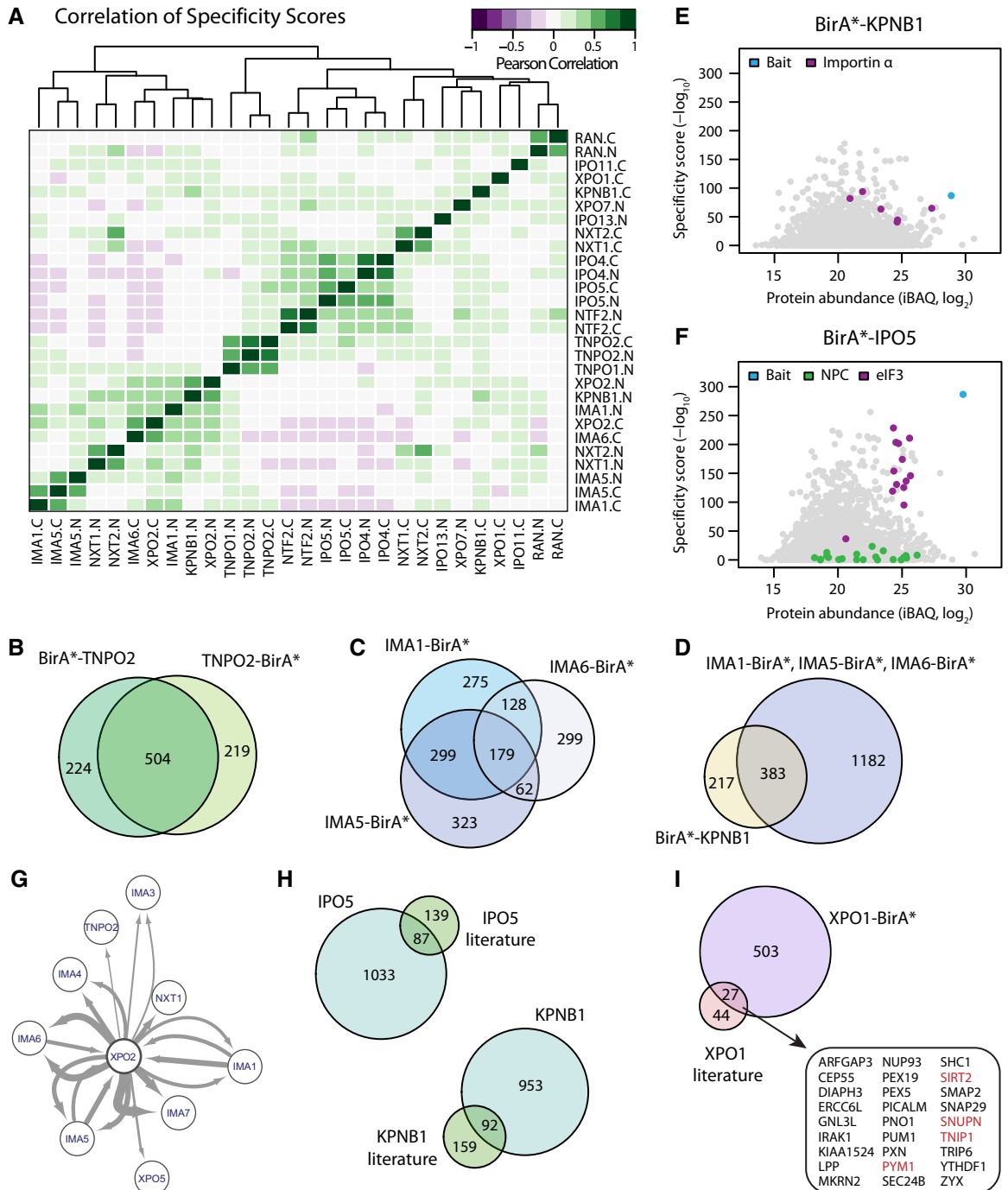

**Figure 4. Specificity and overlap of different transport pathways.**

A–D  (A) Pearson correlation of the specificity scores of all 28 experiments (excluding the four controls). N- and C-terminally tagged cell lines and functionally related NTRs show a high degree of similarity. The overlap of significant identifications of N- and C-terminally tagged TNPO2 (B) between different importin αs (C) and KPNB1 with importin αs (D) is shown as representative examples. Importin αs show a high degree functional redundancy, and a prominent fraction of significant identifications for KPNB1 overlap with hits from importin αs that are adaptors for cargo binding for KPNB1.

E, F  iBAQ and specificity scores in BirA*-KPNB1 and BirA*-IPO5 for selected protein complexes or group of proteins are shown as scatter plots. Common interaction partners like Nups get penalized by specificity score calculation because they interact with multiple NTRs.

G  Interactions of XPO2 with other NTRs recover known properties of the nucleocytoplasmic transport system, including interactions with importin αs. Arrow thickness is proportional to the specificity score of the interaction. Arrow direction indicates bait (source)–prey (target) relationships. Two arrows pointing in the same direction indicate the N- or C-terminally tagged version of the NTR retrieving the same prey.

H, I  Comparison of cargos known from literature for IPO5, KPNB1 (Kimura *et al*, 2017), and XPO1 (Thakar *et al*, 2013; Kırlı *et al*, 2015; Wühr *et al*, 2015). For XPO1, only cargos significant in at least two out of the three previous large-scale studies were considered. Proteins highlighted in red are well-established XPO1 cargos.

XPO1 cargoes such as PYM (Bono *et al*, 2004), TNIP1 (Gupta *et al*, 2000), and SIRT2 (North & Verdin, 2007). We found that cargos of multiple importin βs retrieved by an independent SILAC approach performed on permeabilized cells were also enriched in the corresponding NTR in our data set (Fig 4H and Appendix Fig S2C). However, the overlap varied between NTRs ranging from 38% for IPO5 to 12% for TNPO1. Both in the case of XPO1 and importin βs, those independent experiments were conducted under very different experimental and biological conditions. Additionally, the partial overlap between data sets might be a consequence of the fact that our approach was explicitly designed from a statistical point of view to unravel interactions of specific NTRs. This aspect (i.e., can a given cargo be transported by alternative NTRs?) was only partially addressed by the other studies, and it was not taken into account in the scores used to define cargos.

To demonstrate that our data set is a useful resource and can be mined to derive novel biological information, we attempted to reciprocally validate cargos and the respective transport pathways. For a manually selected subset of initially 12 cargos, we successfully generated stable cell lines and collected mass spectrometry data (Table EV4) from four biological replicates for 10 N- or C-terminal BirA* cargo fusion proteins (Table EV5). In order to identify specific interactions for each cargo, we applied the same approach used for NTRs and compared all the cargo interactomes including BirA* alone and non-engineered cells in a pairwise fashion. The combined analysis yielded significant hits (adj. Fisher *P*-value < 0.01) for six out of 10 fusion proteins. The negative outcome in four cases might occur due to technical limitations, that is, the sensitivity of the experiment might be insufficient for the selected cargo or the fusion protein might interfere with protein complex formation. For five out of the six cargos that yielded significant hits, the reciprocal analysis accurately recovered the NTRs predicted by the large-scale analysis (Table EV5; Figs 5 and EV2). The BioID data additionally recapitulated known interaction of the fusion proteins with the other members of the respective protein complexes *in situ*. We also identified closely related NTRs, which we had not probed for in the large-scale analysis, further underlining the redundancy of the transport system. Only for SRC2, unexpected NTRs were recovered, namely IMA4 and IMA5 by the reciprocal analysis, whereas TPNO1 and TPNO2 were identified by the large-scale analysis (Table EV5). Careful inspection of the data however revealed that the alternative pathways were detected in both data sets although with relatively low score. For two cargos, integrator complex (subunit 1, INTS11) and eukaryotic translation initiation factor 3 (subunit D, EIF3D), we further tested their NTR specificity using gene silencing (Fig 4H). For INTS11, both large-scale and reciprocal analyses had detected importin αs as major NTRs. As expected, gene silencing of these resulted in a decreased nuclear/cytoplasmic distribution ratio as compared to control experiments (Fig 5C and D). Similar results were obtained for importin βs in case of EIF3D (Fig EV2D and E).

The definition of NLSs and NESs of the different NTRs is at present relatively loose and largely based on a very few validated cargos. The automated annotation of NLSs and NESs is challenging and prediction algorithms operate with a limited accuracy, in part because larger training data sets are missing. We therefore wanted to exploit our large-scale data set in this regard. We used cNLS Mapper (Kosugi *et al*, 2009) to predict potential signals in the cargos selected for validation and detected these in INTS11, EXOSC10, and

RPC3 (Table EV6). We also used the DILIMOT algorithm (Neduva *et al*, 2005) and simple regular expression pattern matching to extract short linear motifs in an unbiased way. We used it to detect motifs that are common to proteins that we identified to be significantly enriched for a specific NTR within the large-scale data set. We globally applied this method, although some NTRs, for example, IPO13 (Grünwald *et al*, 2013), recognize folds and not short linear motifs. This analysis recovered KR-rich motifs that are part of the cNLS as the main motif of importin α cargos, and P-rich motifs similar to the PY-NLS as the main motif of transportin cargos (Lee *et al*, 2006), as expected (Table EV6), despite the limitation that DILIMOT is not designed to recover bipartite motifs. In both cases, the identified motifs are similar but not identical to the definitions proposed in previous literature (Kalderon *et al*, 1984; Dingwall *et al*, 1988; Lee *et al*, 2006), underlining that the respective signals exhibit a certain degree of plasticity. Surprisingly, this analysis also identified acidic stretches (DE-rich motifs) to be significantly enriched in cargos of various importin β-type NTRs but not, for example, transportins (Table EV6). Mutational analysis of predicted cNLSs in three out of three of the above-mentioned cargo-BirA* fusion proteins disturbed nuclear localization as expected (Table EV6; Fig 5E and F). We also mutated one of the DE-rich stretches, namely at the C-terminus of EIF3D. The deletion of this motif, but not its substitution of with a neutral linker, led to increased nuclear localization (Fig EV2F and G). The exact physiological role of this potential motif thus needs to be further investigated in the future. Alternatively, it might represent surface properties common to, for example, nuclear or nucleolar proteins.

## Protein complexes and functionally related proteins interact with specific NTRs

In order to characterize the degree of specificity of the NTR-interacting proteome, we used two complementary approaches. First, for each individual NTR, we ranked the interacting proteins according to their specificity score and looked for GO terms enriched in highly specific interaction partners. Our analysis revealed that functionally related proteins involved in similar biological processes tend to show similar interaction patterns with NTRs (Fig 6A). For example, we found the major nuclear biological processes including transcription, chromatin organization, and RNA processing and transport to be associated with importins, but also categories linked to vesicular transport and cytoskeleton organization to be specifically enriched in certain NTRs such as IPO4 and IPO5. These results suggest a coordination of the nucleocytoplasmic transport system based on protein function and indicate that NTRs might be involved in other cellular process beyond their canonical transport function.

In order to support these findings, we used an alternative approach based on network analysis. We asked whether proteins known to interact with each other would show similar patterns of interactions with NTRs. We generated NTR-specific protein networks by applying a network smoothing approach using specificity scores (Appendix Fig S3). Subsequently, we ranked proteins based on the smoothed scores and extracted the top 2% nodes for each NTR sample (309 proteins; Fig 6B). Using this approach, we found in an unbiased way that members of protein complexes show consistent enrichment with specific NTRs. For IMA1-BirA*, many different complexes could be identified like the LSm, exosome, or

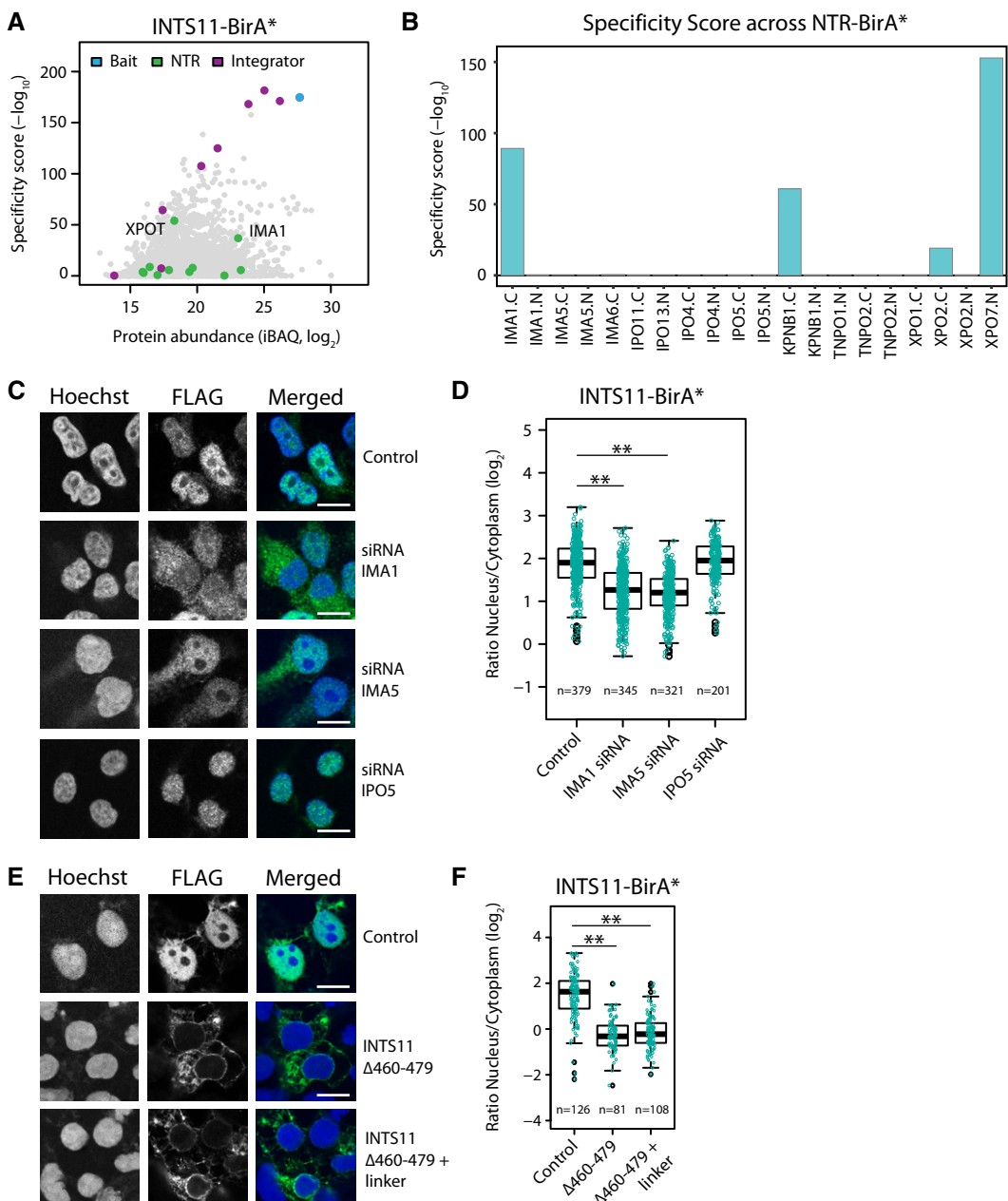

**Figure 5. Small-scale validation of cargos.**

A   iBAQ and specificity scores identified with INTS11-BirA* are shown as scatter plot; members of the integrator complex and NTRs are highlighted.

B   Specificity scores obtained with various NTR-BirA* fusion proteins for INTS11.

C   Subcellular distribution of the INTS11-BirA* upon siRNA treatment against IMA1, IMA5, IPO5, and a negative control (scrambled siRNA). Importin αs induce a shift of INTS11 toward the cytoplasm.

D   Quantification of the ratio of nucleoplasmic to cytoplasmic (N/C) distribution of INTS11 upon siRNA treatment (**Wilcoxon signed-rank test *P*-value < 0.01).

E   Subcellular distribution of INTS11-BirA* upon removal and replacement of the predicted cNLS with a linker. The absence of the predicted cNLS leads to an almost exclusive cytoplasmic localization of INTS11.

F   Quantification of the N/C ratio upon cNLS removal (**Wilcoxon signed-rank test *P*-value < 0.01).

Data information: Scale bar, 10 μm. Boxplots: the upper and lower limit of the box indicate the first and third quartile, respectively, and whiskers extend 1.5 times the interquartile range from the limits of the box. The data for INTS11 and EIF3D (Fig EV2) are shown in an exemplifying manner for all validated cargos (Table EV5).

mediator complex (Fig 6C and D). Multiple members of each complex were retrieved and displayed high specificity scores (Figs 6C and D and, EV3). However, direct biotinylation was observed only for one or few members of each complex. The patterns of biotinylation were extremely reproducible, and similar for interactors shared between different NTRs (note, e.g., the

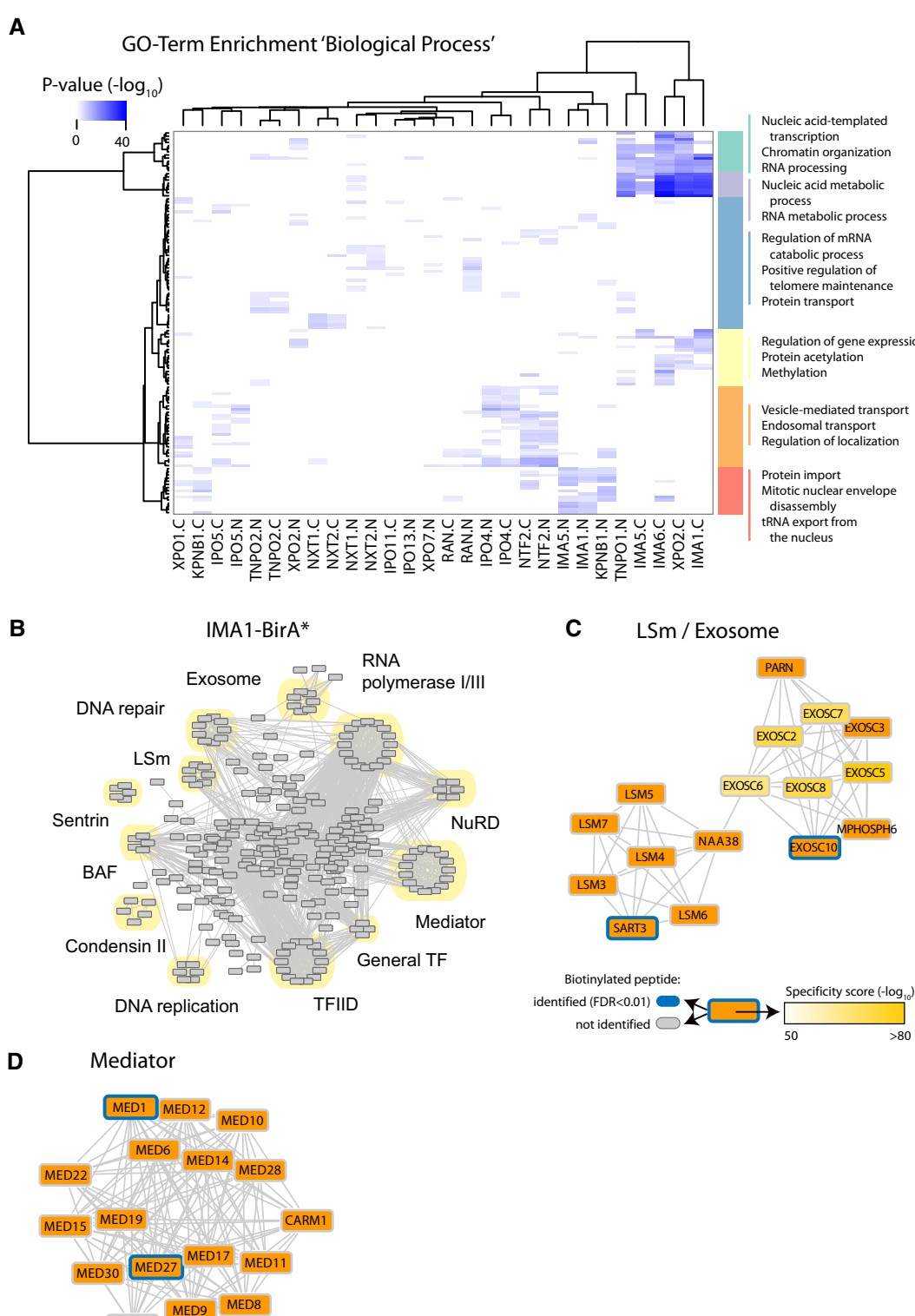

**Figure 6.  Different biological functions und protein complexes are associated with specific NTRs.**

A Comparison of GO terms enriched among proteins interacting specifically with NTRs. GO enrichment was performed individually for each NTR sample by ranking proteins according to their specificity scores. Significant GO terms (*P*-value < 0.001) from the category "biological process" were combined and compared across NTRs. Distinct biological processes display specific association with related NTRs.

B Network analysis of the top 2% enriched proteins of the IMA1-BirA* experiment. Various import cargos are found associated with IMA1. LSm (like Sm), BAF (BRG1- or HBRM-associated factors), NuRD (nucleosome remodeling deacetylase), TF (transcription factor), TFIID (transcription factor II D).

C, D Selected subnetworks are highlighted; specificity scores are indicated by color gradients and detected biotinylated proteins by blue frames.

selective biotinylation of eIF3D and eIF3E by both IPO4 and IPO5; Fig EV3B and C). Similar observations can be made for almost all the other NTRs (see Figs EV3 and EV4 for selected examples). Taken together, these results indicate that many protein complexes are transported across the nuclear envelope as pre-assembled entities, as previously shown for proteasomes (Burcoglu *et al*, 2015) and suggested for other complexes (Hardeland & Hurt, 2006), while only specific members of a given complex establish direct interactions with the respective NTRs.

Using our network approach, we identified several protein complexes that were enriched for specific NTRs (Fig 7A). In agreement with the GO analysis, we found clusters of proteins belonging to the same subcompartment (e.g., nucleolus) and similar biological function (e.g., CREB-related transcription factors) to show specific interaction patterns. Inspired by this finding, we systematically searched for specific relationships between NTRs and transcription factors (TFs). TFs have a strongly varying abundance within the cell, and often they are difficult to identify due to their low copy number. In total, 315 out of the 1,687 TFs defined by the FANTOM5 database (Abugessaisa *et al*, 2016) were contained in our data set and 243 were significantly enriched in at least two distinct experiments (Fig 7B). Highly specific clusters were most often detected in single or highly related NTRs (Table EV7). Our data suggest that despite a certain degree of redundancy between NTRs, selective transport routes exist for specific and functionally related protein complexes and transcription factors. These specificities might underlie the regulatory functions of NTRs in development and disease.

**The direct identification of biotinylated peptides reveals specific interaction sites of NTRs with FG-Nups**

We next investigated the interaction of NTRs with the NPC by taking advantage of the above-introduced direct identification of biotinylation sites (for an overview of all biotinylated peptides identified, refer to Appendix Fig S1E and Table EV8). Our data show a highly specific pattern common to all the NTRs that is characterized by a preferential biotinylation of the peripheral components of the NPC (Fig 8A). The protein abundance (iBAQ score) generally correlated with the amount of biotinylated peptides identified, with some exceptions such as the Nup62 complex (see Discussion). Biotinylated peptides that were identified in scaffold Nups did almost never locate into known structured domains. Similar observations were made for FG-Nups. Structured regions like the RAN-binding domain of NUP50 and NUP358, the beta propeller of NUP214, or the tetratricopeptide repeats (TPR) of NUP358 were hardly ever biotinylated. Rather, the actual NTR-interacting loops and intrinsically disordered FG domains were heavily biotinylated (Figs 8B and C, and EV5).

An overview of all biotinylation sites identified in NUP50, NUP153, NUP98, NUP214, and NUP358 in at least two out of four biological replicates is shown in (Figs 8B and C, and EV5). Some of these sites correlate with known properties of FG-Nups. For example, NUP50 is known to be important for the disassembly of import complexes at the nuclear face of the NPC upon binding of the small GTPase RAN, which is considered to be rate limiting (Güttler & Görlich, 2011). Its N-terminus contains two importin α binding sites (Binding site 1: 1–15; Binding site 2: 24 or 29–46; Ogawa *et al*,

2010). Indeed, the N-terminal sites 59 and 62 display the highest biotinylation intensity across all experiments, while the other biotinylation sites within NUP50 cluster around its FG-repeats (Fig 8B).

Interestingly, FG-Nups also show differential biotinylation patterns. The sites 110, 127, and 229 of NUP50 seemed to be highly favored by importin αs and transportins. In contrast, IPO4, IPO5, NXT1, NXT2, and the exportins, except XPO2, did almost never biotinylate any of these sites. Importin βs (except for KPNB1 that binds to importin αs) show very little interactions with nuclear basket protein NUP153 (Fig 8C) but heavily modify the cytoplasmic filament protein NUP358 (Fig EV5C), whereas importin αs and transportins have opposite preferences. At last, NXTs and to some extent exportins seem to prefer the N-terminal domain of NUP153 over its C-terminal FG-domain, while this is not true for transportins and importin αs. These data strongly suggest preferred interaction sites of specific NTRs *in situ* and offer an attractive explanation for why there are different types of FG-Nups. In summary, our approach provides a high level of detail about the interaction of NTRs with Nups *in situ*. This information can be used to design future experiments aimed at investigating specific relationships between NTRs and Nups.

## Discussion

Here, we report a comprehensive picture of the landscape of nuclear transport receptor specificity. Methodologically, our analysis complements previous work that relied on alternative approaches. The major advantage of *in situ* proximity ligation is that it is done in living cells and thus preserves all subcellular structures, protein concentrations, and regulatory networks (Hung *et al*, 2014; Coyaud *et al*, 2015), while several previous studies relied on protein extraction or permeabilized cells. Nevertheless, cross-validation against such previous data showed that similar set of cargos are identified (Kimura *et al*, 2017). Most importantly, we introduced a statistical framework for the BioID method that allows to quantify the specificity of cargos for individual NTRs and to determine the FDR for our identifications.

Our data set comprises a rich resource containing four layers of information. The raw data (Table EV4) contain iBAQ scores that are representative for the enrichment of the identified proteins within a particular experiment. These data are not corrected for contaminants, such as naturally biotinylated proteins. The second layer is the NIP (Table EV2). This set contains proteins that are significantly enriched in our experiments as compared to controls and is corrected for background identifications. It contains many high abundant cargos including those that are not associated with a specific NTR. This could be because either the respective NTR was not analyzed in our study or because they utilize multiple NTRs. The third layer is based on specificity scores that were designed to discover proteins that exclusively interact with one or very few NTRs (Table EV4). Here, proteins identified in multiple experiments such as Nups, RAN, or cargos that utilize multiple NTRs are penalized. These three layers are based on indirect identification of peptides eluted after on-bead digestion, and they are represented on the protein level, meaning that the signal from all peptides per protein was integrated. Complementary to these, we

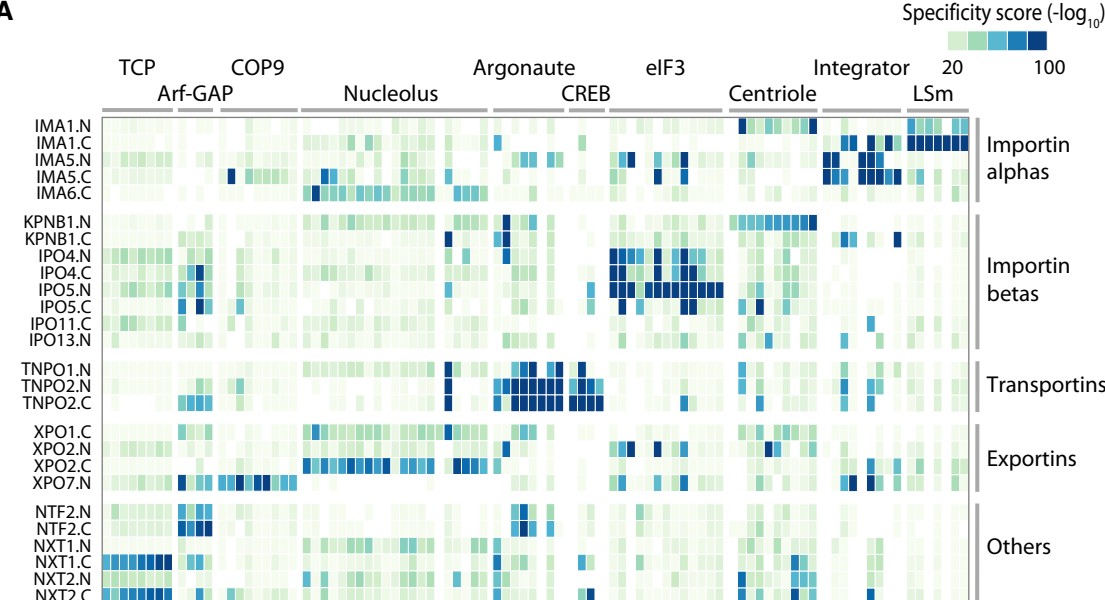

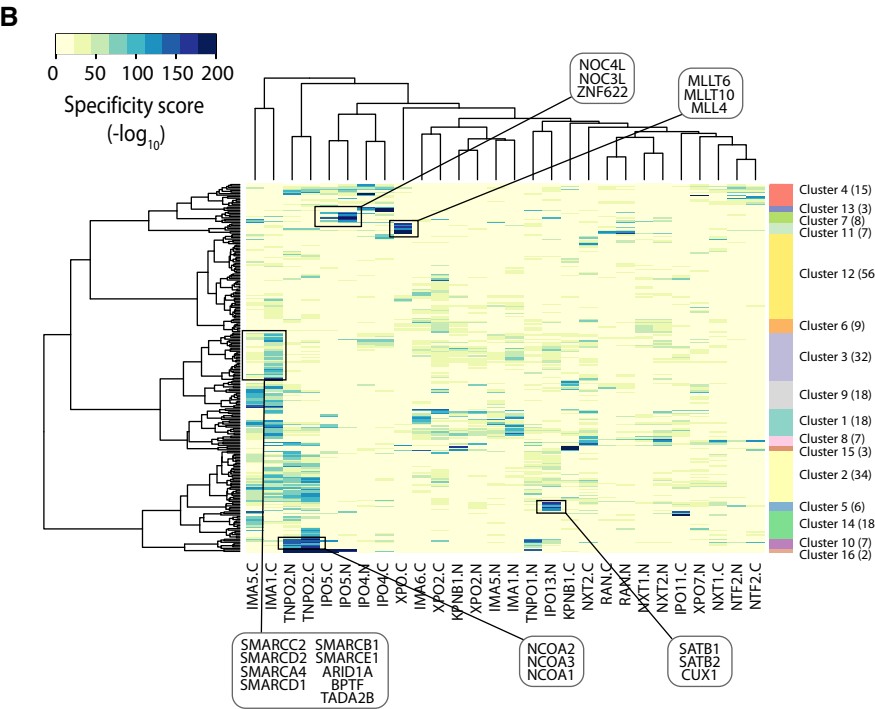

**Figure 7. Transcription factors and regulatory protein complexes are associated with specific NTRs.**

A   Specificity scores of selected protein complexes and functionally related proteins are shown as a heat map across all experiments. Multiple subunits of the depicted protein complexes are identified to interact with specific NTRs [TCP (=CCT chaperonin containing TCP-1), ArfGAP (ADP-ribosylation factor GTPase-activating proteins), COP9 (constitutive photomorphogenesis 9), CREB (cAMP response element-binding protein), eIF3 (eukaryotic initiation factor 3)]. Immunofluorescence staining supports preferential biotinylation of nucleolar proteins by certain NTRs (Fig 2B).

B   Cluster analysis of transcription factors (TF) significantly enriched (adj. Fisher P-value < 0.01) in at least two experiments are shown. Selected clusters and related TFs therein are highlighted. Related TFs use the same or similar NTR.

provide identification of biotinylation sites obtained by separate elution and analysis of biotinylated peptides, which are represented on the peptide level (Table EV8). The number of sites

identified in a protein strongly correlates with known interactions. The sites are identified much more frequently in intrinsically disordered proteins (IDPs) as compared to folded domains and in case

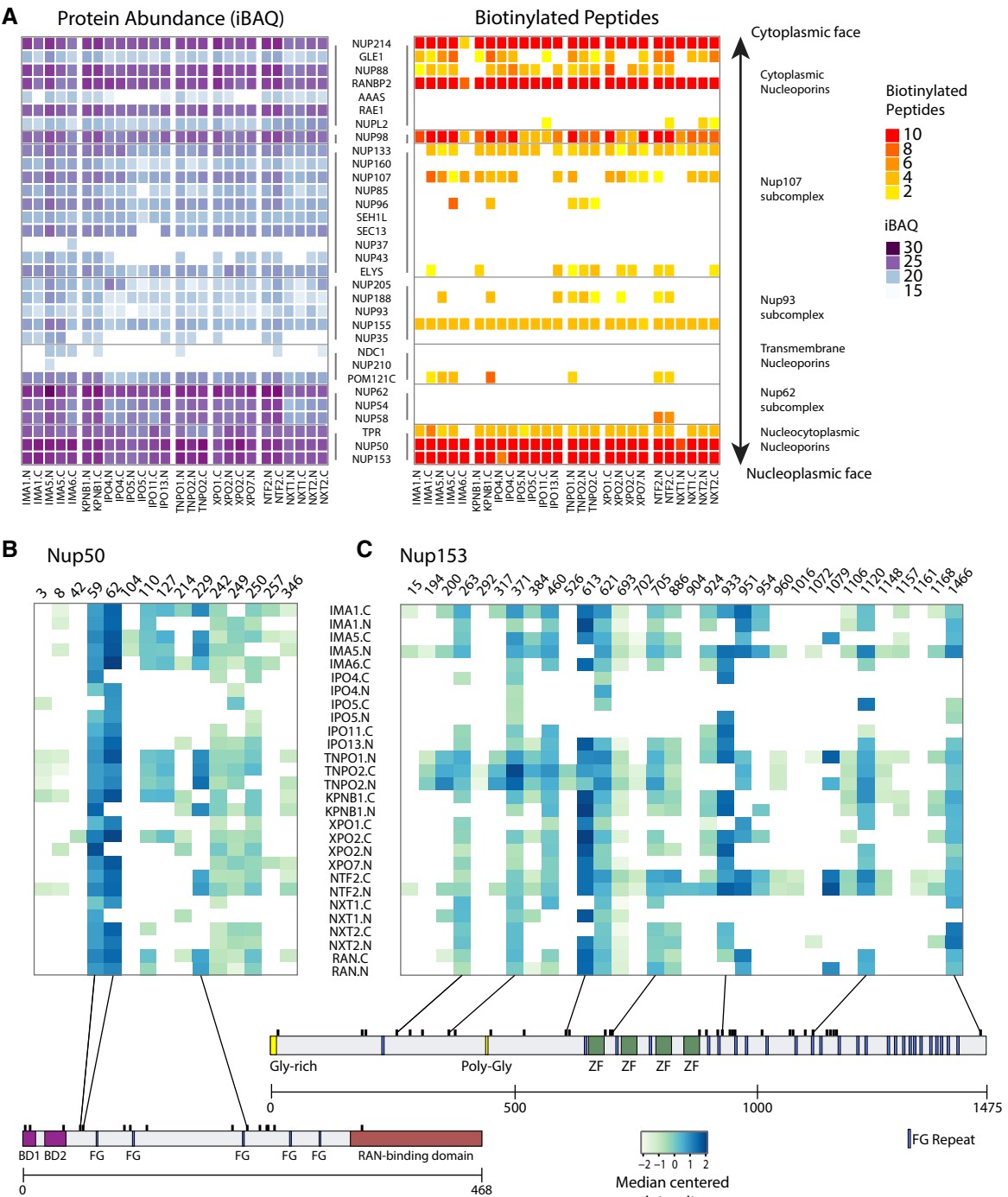

**Figure 8.  Direct identification of biotinylated peptides suggests that NTRs preferentially bind specific FG-NUPs *in situ*.**

A     NTRs interact with specific structures of the NPC. Protein abundances (iBAQ scores) of Nups across NTR samples often correlate with the number of identified biotinylated peptides.

B, C   Normalized and median centered intensities of biotinylated peptides of NUP50 and NUP153. Structural domains like the importin α binding site (BD1 and BD2), RAN-binding domain, zinc fingers, and FG-repeats are indicated. IPO4, IPO5, and IPO11 show less pronounced intensities as compared to Nup358 and Nup214 (Fig EV3).

of multidomain proteins might also be explored for identifying binding sites. Within protein complexes, usually all subunits are identified jointly by the indirect method, but only very few

subunits are directly biotinylated. The data might thus also be further explored to identify subunits that harbor yet unknown NLSs or NESs in order to discriminate piggy back translocations of

**Table 1.    Overview of enriched protein complexes and identified biotinylated subunits.**

| NTR | Enriched protein complexes and proteins of similar biological function | Number of identified subunits (ES > 25) | | Biotinylated subunit |
| | | NTR-BirA* | BirA*-NTR | |
| --- | --- | --- | --- | --- |
| IMA1 | Condensin ll complex | 5 (5) | 4 (0) | SMC4, hCAP-D3 |
| | H/ACA ribonucleoprotein complex | 5 (5) | 3 (0) | DKC1 |
| | DNA replication complex GINS | 3 (3) | 1 (0) | – |
| | Little elongation complex | 4 (4) | 4 (3) | ELL, ICE1 |
| | Integrator complex | 8 (6) | 7 (0) | INTS4, INTS10 |
| | Exosome complex | 11 (10) | 11 (0) | EXOSC9, EXOSC10, RRP44 |
| | DNA-directed RNA polymerase l | 9 (9) | 7 (5) | – |
| | DNA-directed RNA polymerase lll | 15 (14) | 11 (7) | RPC3 |
| | Chromodomain-helicase-DNA-binding protein | 6 (5) | 6 (1) | CHD4, CHD7, CHD8 |
| | DNA mismatch repair protein | 5 (5) | 5 (2) | MLH1, MSH6 |
| | Mediator of RNA polymerase II transcription | 19 (17) | 13 (3) | MED1, MED27 |
| | MORC family CW-type zinc finger protein | 3 (3) | 3 (1) | – |
| | Negative elongation factor | 4 (4) | 4 (1) | NELF-A, NELF-E |
| | Transcription initiation factor TFIID | 13 (13) | 11 (11) | TAF1, TAF3, TAF7 |
| | U6 snRNA-associated Sm-like protein | 7 (7) | 6 (6) | – |
| | Putative polycomb group protein ASXL | 3 (3) | 2 (0) | – |
| IMA5 | Condensin ll complex | 4 (1) | 2 (0) | SMC4 |
| | H/ACA ribonucleoprotein complex | 5 (5) | 5 (4) | DKC1 |
| | Little elongation complex | 4 (3) | 4 (2) | ELL |
| | Integrator complex | 9 (8) | 8 (5) | INTS4, INTS10, INTS11 |
| | Exosome complex | 10 (6) | 11 (1) | EXOSC9, EXOSC10, RRP44 |
| | DNA-directed RNA polymerase l | 9 (7) | 7 (5) | RPAC2 |
| | DNA-directed RNA polymerase lll | 13 (9) | 10 (5) | RPAC2, RPC3 |
| | Chromodomain-helicase-DNA-binding protein | 6 (5) | 5 (2) | CHDL1, CHD3, CHD7, CHD8 |
| | DNA mismatch repair protein | 4 (3) | 5 (1) | MLH1, MSH6 |
| IMA6 | H/ACA ribonucleoprotein complex | 5 (4) | – | DKC1 |
| | Little elongation complex | 4 (3) | – | ICE1 |
| IPO4 | Anaphase-promoting complex | 9 (3) | 9 (4) | APC3 |
| | AP2 adaptor complex | 5 (3) | 4 (1) | – |
| | HAUS augmin-like complex | 8 (8) | 8 (7) | HAUS2* |
| | Eukaryotic translation initiation factor 3 | 13 (9) | 13 (11) | Subunit D, E, G, J |
| | Activating signal cointegrator (ASC-1) | 4 (4) | 4 (4) | TRIP4, ASCC1 |
| | Histone chaperone ASF1 | 2 (2) | 2 (2) | – |
| | Transforming acidic coiled-coil-containing protein | 3 (3) | 3 (3) | TACC1, TACC3 |
| IPO5 | Anaphase-promoting complex | 8 (0) | 9 (9) | – |
| | AP2 adaptor complex | 5 (0) | 4 (4) | – |
| | HAUS augmin-like complex | 8 (2) | 8 (7) | – |
| | Eukaryotic translation initiation factor 3 | 12 (6) | 13 (13) | Subunit A, D, E, G, J |
| | WASH complex | 4 (2) | 4 (4) | – |
| IPO11 | Basic leucine zipper and W2 domain-containing protein | 2 (2) | – | BZW1, BZW2 |
| IPO13 | Special AT-rich sequence-binding protein | – | 2 (2) | SATB1 |
| TNPO1 | Argonaute protein family | – | 2 (2) | – |
| | Nuclear receptor coactivator | | 6 (5) | SRC2* |

                  

**Table 1** (continued)

| NTR | Enriched protein complexes and proteins of similar biological function | Number of identified subunits (ES > 25) | | Biotinylated subunit |
| | | NTR-BirA* | BirA*-NTR | |
|---|---|---|---|---|
| TNPO2 | Argonaute protein family | 3 (3) | 3 (3) | – |
| | Nuclear receptor coactivator | 6 (6) | 6 (6) | – |
| | CREB-regulated transcription coactivator and binding protein | 4 (4) | 4 (4) | CRTC1, CRTC2, CRTC3 |
| | Far upstream element-binding protein | 3 (2) | 3 (3) | FUBP1, FUBP2, FUBP3 |
| | CCAAT-box-binding transcription factor (CTF) | 3 (3) | 3 (2) | – |
| XPO1 | Little elongation complex | 4 (4) | – | – |
| | Exosome complex | 10 (2) | | EXOSC10 |
| XPO2 | DNA replication complex GINS | 3 (3) | 2 (1) | – |
| | Heterogeneous nuclear ribonucleoprotein | 11 (0) | 21 (17) | Protein A1, F, M, U, UL1 |
| XPO7 | COP9 signalosome | – | 9 (9) | CSN4 |

Proteins labeled with an asterisk were only identified in the ACN/TFA elution dataset.

protein complex subunits from direct NTR binding mechanisms of individual proteins.

A disadvantage of the BioID system is that molecular tags have to be introduced. In case of importins and exportins, previous work has shown they can be functionally tagged (Miyamoto *et al*, 2002; Ciciarello *et al*, 2004; Kimura *et al*, 2013b; Vuković *et al*, 2016). This is very well reflected in our data, for example, biotinylated cargos accumulate in the expected cellular compartment over time and known interactors are among the most prominently identified proteins. An exception might be the N-terminal importin β binding domain (IBB) of IMA1 and IMA5 that is required for the binding of these adaptor proteins to importin β, and could be sterically hindered by BirA* tagging. Indeed, the N-terminally tagged lines for these two NTRs behave somewhat as outliers in our data set: Biotin stainings are faint and the number of identified proteins is much lower, indicating that the accumulation of cargos over time works less well. Also the observed correlation with the C-terminally tagged lines is only moderate in these two cases (Fig 4A). For transparency, we nevertheless decided to publish these data together with the entire set. The independent analysis of N- and C-terminal tagged lines is therefore an important validation that we have done for the majority of all NTRs investigated here. One also might caution the interpretation of the auxiliary NTRs in the data set, specifically RAN and NTF2 since the tag might have a more severe effect onto these rather small proteins. In case of RAN, GFP fusions show accurate subcellular localizations throughout the cell cycle (Hutchins *et al*, 2009). Known interactors of RAN are among the top identified proteins in our data set. This includes XPO5, XOP7, RanGNRF, its deacetylase sirtuin-2 (de Boor *et al*, 2015), IPO9, XPO1, RANBP1, and IPO7 that all are among the top 20 identified proteins in this order. We believe that the RAN data sets will also contain many export cargos of NTRs not targeted in this study because RAN is part of the respective export complexes and interacted with various exportins in our data. Among the most prominent proteins identified with direct biotinylation sites in the same experiments are NUP358/RANBP2, NUP214, NUP153, NUP50, RANBP1, and RANGAP1 indicating that RAN binds to Nups on both faces of the NPC, as expected. The aforementioned finding also shows that indirect and direct identification of biotinylated proteins and peptides,

respectively, complement each other. Another example for this is the NUP62 complex, a very prominent FG-Nup complex within the NPC's central channel. All three members of this subcomplex are found in high abundance in basically all indirect experiments, but direct biotinylations sites are rare. We obtained similar results for the FG-repeat domains at the N-terminus of NUP98 and C-terminus of NUP214 (Fig EV5A and B). This behavior might indicate a more stochastic or transient binding of cargo complexes. Alternatively, biophysical properties of this complex might be the cause. As a matter of fact, tryptic cleavage sites are somewhat reduced in the respective FG domains of all five proteins.

Previous studies of XPO1 relied on a very potent inhibitor (leptomycin B; Thakar *et al*, 2013; Wühr *et al*, 2015), or the fact that export complexes are formed in the presences of RAN-GTP for cargo identification (Kırlı *et al*, 2015). Cross-validation of the cargos identified in our study against previous work on XPO1 showed statistically significant, but incomplete overlap (Fig 4I). We believe that this apparent discrepancy is rather funded in the biology of XPO1. Its key function is the clearance of proteins that "leaked", potentially even unspecifically into the nucleus, back into the cytoplasm. Since previous work has been done in very different cell types, for example, *Xenopus* oocytes, we believe it is conceivable that the set of proteins transported under such conditions is different. In line with this view, and in strong contrast to other importins and exportins, the cargos identified for XPO1 cluster to a much lesser extend into protein complexes that would have a smaller chance to "leak" across the permeability barrier. This is contrasted by XPO2 and XPO7 for which we obtained broad data sets (Fig EV4). XPO2 has been thought to specifically recycle importin αs back into the cytoplasm, and it has been proposed that this has to be the case because nuclear XPO2 antagonizes importin α cargo recognition once the cargo has been released into the nucleoplasm (Güttler & Görlich, 2011). In our hands, XPO2 indeed interacted with six out of seven human importin αs (Fig 4G), but also a large set of other proteins, even more than XPO1. We reasoned that this might occur because XPO2 biotinylates these proteins when it encounters importin α-β-cargo complexes upon their disassembly at Nup50. We therefore compared cargos identified in experiments with importin αs to XPO2. There was indeed a considerable overlap between the data

and some obvious import cargos, such as the spliceosome, PCAF, or histones were identified with XPO2 (Fig EV4B). This illustrates that the interactome derived from proximity ligation covers indirect interactions and it should be taken into account when the data are explored. Very little was previously known about XPO7 that interacted with a very defined set of protein complexes such as the COP9 signalosome and AMPK (Fig EV4C) and showed an overlap with IPO4 and IPO5 (Appendix Fig S2A).

Our data identify the potential transport pathways for many prominent cargos (Table 1), that were to the best of our knowledge previously unknown such as anaphase-promoting complex (IPO4, IPO5), condensin 2 (IMA1), H/ACA ribonucleoprotein complex (IMA1, IMA5), HAUS augmin-like complex (IPO4, IPO5), argonaute (TNPO2), COP9 (XPO7), DNA replication complex GINS (IMA1, XPO2), little elongation complex (XPO1), integrator complex (IMA5), exosome complex (IMA1, IMA5, XPO1), as well as nuclear receptor coactivator (NCoA; TNPO2, TNPO1), and various other transcription factors (Fig 7B). This list includes a number of cargos that were previously difficult to assign to a specific transport pathway, such as CREB (Ch'ng *et al*, 2015; TNPO1, TNPO2), AP2 complex (Kitagawa *et al*, 2008; IPO4, IPO5), and RNA polymerase III (Hardeland & Hurt, 2006; Appendix Fig S4). We believe that an even larger set of cargos for all NTRs could be identified in the future if our approach will be applied to other cell types and biological conditions.

# Materials and Methods

## Human cell culture

Flp-In 293 T-REx (Thermo Fisher Scientific) and HEK (human embryonic kidney) cells were grown in Dulbecco's modified Eagle's medium (DMEM) high glucose 5 g/l (Sigma-Aldrich) supplemented with 10% heat inactivated fetal bovine serum (FBS). The parental cell line was grown with the addition of 100 μg/ml Zeocin (Invitrogen) and 15 μg/ml blasticidin (Thermo Fisher Scientific). After generation of stable cell lines, Zeocin™ was replaced by 100 μg/ml hygromycin. Stable cell lines were seeded at a density of $1.6e4$ cells/cm², allowed to attach to the culture dish for 24 h, and then induced by adding 1 μg/ml tetracycline (Sigma-Aldrich) in ethanol directly to the medium. After 24 additional hours, 50 μM biotin (Sigma-Aldrich) of a 10 mM stock solution in water was added.

## NTR-BirA* and cargo-BirA* plasmid and stable cell line generation

For the generation of plasmids for genomic integration into the Flp-In T-Rex cells, the Gateway Technology (Invitrogen) was used. Expression clones were generated by combining a destination vector (pcDNA5-pDEST-BirA*-FLAG-N-term or pcDNA5-pDEST-BirA*-FLAG-C-term) with an entry clone. The following entry clones were purchased from the human ORFeome collection either as cDNA or as entry clone: IMA1 (BC005978), IMA5 (BC002374), IMA6 (BC047409), KPNB1 (BC003572), IPO4 (BC136759), IPO5 (BC001497), IPO11 (BC033776), IPO13 (BC008194), TNPO1 (BC040340.1), TNPO2 (BC072420), XPO1 (BC032847), XPO2 (BC109313), XPO7 (BC030785), RAN (BC014901), NTF2 (BC002348.2), NXT1 (BC003410), NXT2

(BC014888.1), APC2 (BC011656), CSN4 (BC093007), DKC1 (BC009928), EIF3D (BC093686), EXOSC10 (BC073788), HAUS2 (BC010903), INTS11 (BC007978), RPAC2 (BC000889), RPC3 (BC002586), SRC2 (BC114383). Entry clones were generated using the donor vector pDONR. Purchased entry clones or cDNAs with sequence disagreements with current sequences available at ENSEMBL were changed using the QuickChange II Site-Directed Mutagenesis Kit (Agilent). Required attB sites to generate an entry clone were added by PCR using the Phusion High-Fidelity DNA Polymerase (Thermo Fisher Scientific). Stable cell lines were generated using X-tremeGENE 9 DNA Transfection Reagent (Sigma-Aldrich).

## BioID affinity purification (AP)

For each experiment, $4e7$ snap-frozen cells were used. The AP was performed as previously described (Coyaud *et al*, 2015). Instead of a protease inhibitor mixture, 1 mg/ml aprotinin and 0.5 mg/ml leupeptin was used. 1 μg of trypsin (Mass Spectrometry Grade, Promega) was added and incubated at 37°C for 16 h shaking at 500 rpm. Subsequently, 0.5 μg of trypsin was added and the on-bead digest continued for additional 2 h. The beads were transferred to a spin column, and the digested peptides were eluted with two times 150 μl of 50 mM ammonium bicarbonate. To remove the biotinylated peptides still bound to the beads, 150 μl of 80% ACN and 20% TFA was added, briefly mixed, and eluted; this step was done twice. The ACN/TFA elutions were merged. The samples were dried using a vacuum centrifugation. The elutions were resuspended in 200 μl buffer A. The desalting and clean-up of the samples were carried out using Micro Spin Columns (Harvard Apparatus).

## Protein identification by mass spectrometry and label-free quantification

The shot-gun MS experiments were performed as previously described (Mackmull *et al*, 2015). The elutions of biotinylated peptides were measured using the same settings but with a stepwise gradient lasting 90 min. For the quantitative label-free analysis, raw files from a Orbitrap Velos Pro instrument (Thermo) were analyzed using MaxQuant (version 1.5.3.28; Cox & Mann, 2008). MS/MS spectra were searched against the *Human* Swiss-Prot entries of the UniProt KB (database release 2016_09, 19,594 entries) using the Andromeda search engine (Cox *et al*, 2011). The protein sequences of BirA* and streptavidin were added to the database. The search criteria were set as follows: Full tryptic specificity was required (cleavage after lysine or arginine residues, unless followed by proline); three missed cleavages were allowed; oxidation (M), acetylation (protein N-term), and biotinylation (K) were applied as variable modifications, if applicable, mass tolerance of 20 ppm (precursor) and 0.5 Da (fragments). The retention times were matched between runs, using a time window of 3 min. The reversed sequences of the target database were used as decoy database. Peptide and protein hits were filtered at a false discovery rate of 1% using a target-decoy strategy (Elias & Gygi, 2007). Additionally, only protein groups identified by at least two unique peptides were retained. The intensity per protein of the proteinGroups.txt output of MaxQuant was used for further analysis. All comparative analyses were performed using R version 3.2.2. (R Core Team, 2012). The R packages MSnbase (Gatto & Lilley, 2012) was used for processing

proteomics data, and the included package imputeLCMD was used for imputing missing values based on the definitions for missing at random (MAR) and missing not at random (MNAR) values. MNAR were defined for each pairwise comparison as values that were (i) missing in four out of four, or three out of four biological replicates in one sample group, and (ii) present in at least three out of four biological replicates in the second sample group. Because of their non-random distribution across samples, these values were considered as underlying biological difference between sample groups. MNAR values were computed using the method "MinDet" by replacing values with minimal values observed in the sample. MAR values were consequently defined for each pairwise comparison as values that were missing in one out of four biological replicates per sample group. MAR values were imputed based on the method "knn" (k-nearest neighbors; Gatto & Lilley, 2012). All the other cases (e.g., protein groups that had two or fewer values in both sample groups) were filtered out because of the lack of sufficient information to perform robust statistical analysis. The data were quantile normalized to reduce technical variations (Gatto & Lilley, 2012). Protein abundance variation was evaluated using the Limma package (Smyth *et al*, 2005). Differences in protein abundances were statistically determined using the Student's *t*-test (one-sided) with variances moderated by Limma's empirical Bayes method.

The pairwise comparisons of the AP of NTRs to the control data set were used to define the NIP and background proteome. The comparisons were done one-sided and separately for the NIP and the background proteome. The Sime's adjusted *P*-values were calculated for each protein using the R cherry package (Goeman & Solari, 2011). The minimum Sime's *P*-value per protein, which defines if this protein is significant in the NIP or background proteome, was adjusted using the method of Benjamini and Hochberg. This idea was adapted from (Lun & Smyth, 2014): The minimum Sime's *P*-value will be small if the protein of interest is truly significantly differentially abundant in at least one of the comparisons to the control conditions under consideration.

The specificity score per protein in each experiment was calculated by multiplying all significant *P*-values (*P*-values < 0.01) of the selected protein obtained in the pairwise comparisons (*P*-values of those pairwise comparisons where the protein was identified as enriched). Subsequently, an average adjusted fold change (FC) was calculated by using all FC corresponding to the previously considered *P*-values. The Fisher transformation was used to define a *P*-value per experiment. The Fisher *P*-values were adjusted using the method of Benjamini and Hochberg.

## Network analysis

In order to identify clusters of proteins displaying specific interaction with NTRs, we applied a network propagation approach similar to the one described in (Vanunu *et al*, 2010). First, we mapped all the proteins quantified in our experiments to the human STRING protein–protein interaction network (v10, combined score > 0.7, 15,478 nodes). The network was then converted into an adjacency matrix and normalized using Laplacian transformation. Specificity scores where propagated to adjacent nodes by network propagation using the sharing coefficient (α) of 0.5 and 30 iterations. We observed that the standard network propagation algorithm suffered from gene-specific biases created by their network neighborhood ("topology bias"). For example, genes with many neighbors will generally tend to accumulate higher scores independent of their initial specificity scores. Therefore, we devised an additional step of topology bias-correction after the standard network propagation (Appendix Fig S3). We computed each node's topology bias by applying the mean initial score from each sample to all the nodes in the network and then propagating scores using the same parameters (α = 0.5 and iterations = 30). If there was no topology bias, all nodes should have the same scores after this procedure. Resulting propagated scores were used as correction factors for each node and thus subtracted from the original propagated scores. For each NTR sample, proteins were ranked according to their smoothed, topology bias corrected scores, and the top 2% proteins for each sample were used for the identification of highly interconnected subnetworks using the Cytoscape (Cline *et al*, 2007) App MCODE (Bader & Hogue, 2003).

## Gene Ontology enrichment analysis

Gene Ontology enrichment analysis was performed on ranked list of proteins using log-transformed *P*-values or specificity scores using GOrilla (Eden *et al*, 2009) followed by GO term redundancy reduction performed by REVIGO (Supek *et al*, 2011).

## Staining of NTR-BirA* cell lines

Cells were grown directly on glass slides, previously coated with poly-lysine, in PBS for 4 h. The cells were induced and treated with biotin as described above. Between each of the incubation steps at room temperature, the glass slides were washed twice with PBS. First, the cells were fixed with 2% PFA in PBS for 15 min and then permeabilized with 0.4% triton in PBS for additional 15 min. Blocking was performed using 2% BSA and 2% FBS in PBS for 1 h. To visualize the nuclear envelope, cells were incubated with anti-FLAG (1:500, Sigma-Aldrich, #F1804) in blocking buffer for 1 h. As a secondary antibody, an anti-mouse conjugated to Alexa Fluor 488 (1:1,000, Life Technologies, #A21204) was used also for 1 h. All following steps were done with minimum light exposure. Streptavidin covalently bound to Alexa Fluor 647 (1:1,000, Thermo Fisher Scientific, #S21374) in 0.1% BSA in PBS was used to incubate the cells for 10 min. To preserve the stained cells, all glass slides were mounted upside down on a microscope slide using one drop of mounting medium (Thermo Fisher Scientific), dried over night at room temperature, and afterward stored at −20°C.

## Depletion of NTRs by siRNA

Cells were allowed to attach to the dish for 24 h and then transfected with IMA1-specific siRNA (#s7922), IMA5-specific siRNA (#s223980), IPO11-specific siRNA (#s27652), IPO4-specific siRNA (#s36154), IPO5-specific siRNA (#s7935), TNPO2-specific siRNA (#s26880), GAPDH-specific siRNA (#4390849), and negative control no. 1 siRNA (#4390843) purchased from Ambion by Life Technologies. 25 pmol of siRNA (final concentration of 10 nM) was transfected using 7.5 μl of lipofectamine RNAiMAX (Thermo Fisher), according to the manufacturer's protocol. The cells were incubated with the different siRNAs for 72 h. Each treatment was performed in three biological replicates.

## Quantitative PCR (qPCR)

Total RNA was extracted using the RNeasy Plus Mini Kit (Qiagen) following the manufacturer's protocol. For cDNA synthesis, the QuantiTect Reverse Transcription Kit (Qiagen) was used. Quantitative real-time PCR (qRT–PCR) was used to examine the relative expression of IMA1 (5′-ttatcctggatgccatttcaa-3′, 5′-agcctccacattcttcaa tca-3′), IMA5 (5′-gctagtactgtgccgcttcc-3′, 5′-gcaggtacagattgcagtcatc-3′), IPO11 (5′-caaacggtttccatggatct-3′, 5′-ctgtgtctcccactgcttca-3′), IPO4 (5′-cacctctcagcccagttca-3′, 5′-ctcagggacagccctgtaag-3′), IPO5 (5′-tgggaca gatggctacagatt-3′, 5′-acgttgattgccttggtctt-3′), TNPO2 (5′-atcctggatggcaa caagag-3′, 5′-ttcccaaaggcaaagacaag-3′), and normalized to GAPDH (5′-ggtctcctctgacttcaaca-3′, 5′-agccaaattcgttgtcatac-3′). For qRT–PCR analysis, 25 ng of cDNA was used in a 20-μl reaction consisting of 11 μl of SYBR® Green PCR Master Mix (Applied Biosystems), 10 μM forward and reverse primer, and water. Thermocycling was carried out using the StepOne™ (Applied Biosystems) and each sample was measured in technical duplicate. Relative mRNA levels were calculated using the $\Delta\Delta C_t$ method (Livak & Schmittgen, 2001). Significant changes were assessed by applying a Welch two sample *t*-test on the $\Delta C_t$ values for treatment and control samples (Yuan *et al*, 2006).

## Mutation of motifs

Predicted motifs by DILIMOT or cNLS Mapper were removed using the Q5 Site-Directed Mutagenesis Kit [New England Biolabs (NEB)] following the manufacturer's protocol and replaced with a flexible linker (5′-ggtggcggaggtagcggaggcggtggatcg-3′). Transient transfected cells were generated using X-tremeGENE 9 DNA Transfection Reagent (Sigma-Aldrich).

## Image analysis

Image analysis was performed using CellProfiler (Carpenter *et al*, 2006). Nuclei were segmented using CellProfiler's automated maximum correlation thresholding (MCT) algorithm (Padmanabhan *et al*, 2010). To ensure that the segmented nuclei only cover pixels inside the nucleoplasm, the nuclear masks were shrunk by 3 pixels. Next, a ring of 20 pixel width around each nucleus was generated marking the cytoplasm. The area covered by both the cytoplasm and the nucleus was termed cell. The mean intensity of the protein was measured in the cell, nucleus, and cytoplasm region. The measured intensity in the cell was used to filter out cells that had too low protein expression (< 0.08) in the transient transfected cells. The ratio of the mean intensities in the nucleus and cytoplasm was the final readout of the analysis.

## Data availability

The data set generated in this study is available in the following database:

- Mass spectrometry proteomics data: PRIDE PXD007976 (https://www.ebi.ac.uk/pride/archive/projects/PXD007976) (Vizcaino *et al*, 2013).

**Expanded View** for this article is available online.

## Acknowledgements

We thank Drs. Edward Lemke for critical reading of the manuscript, Anne-Claude Gingras, Brian Raught, Jan Ellenberg and Naoko Imamoto for reagents, Laurent Gatto for advice on proteomic data analysis, and Ivana Nikić-Spiegel for help with confocal microscopy. We gratefully acknowledge support from EMBL's advanced light microscopy (ALMF) and proteomic (PCF) core facilities, the FLI functional genomics facility, and Christian Tischer for additional image analysis support. The FLI is a member of the Leibniz Association and is financially supported by the Federal Government of Germany and the State of Thuringia. M.B. acknowledges funding by EMBL and the European Research Council (309271-NPCAtlas). A.B. acknowledges funding by the German Federal Ministry of Education and Research (BMBF; grants: Sybacol & PhosphoNetPPM).

## Author contributions

M-TM, AO and, MB conceived the project, designed experiments, analyzed data, and wrote the manuscript. M-TM, IH, and AO performed experiments. RBR, BK, MC, and AB analyzed data. AO and MB oversaw the project.

## Conflict of interest

The authors declare that they have no conflict of interest.

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
