## [Review Process File · Molecular Systems Biology]

Landscape of nuclear transport receptor cargo specificity

Marie-Therese Mackmull, Bernd Klaus, Ivonne Heinze, Manopriya Chokkalingam, Andreas Beyer, Robert B Russell, Alessandro Ori, Martin Beck

Review timeline:

Submission date:	3 March 2017
Editorial Decision:	6 April 2017
Revision received:	18 August 2017
Editorial Decision:	5 October 2017
Revision received:	23 October 2017
Accepted:	10 November 2017

Editor: Thomas Lemberger

Transaction Report:

1st Editorial Decision

6 April 2017

Thank you again for submitting your work to Molecular Systems Biology. We have now heard back from the referees who agreed to evaluate your manuscript. As you will see from the reports below, the referees find the topic of your study of potential interest. They raise, however, several important points on your work, which should be convincingly addressed in a major revision of the present work.

The reviewers find the data of high quality and the approach interesting. They are however suggesting several further analyses to deepen the insights gained from the study. While we do NOT think that it would be realistic to ask repeating the entire analysis in a second cell line, some additional verifications of the reported interactions and effects on transport and further insights into the redundancy among beta-importing and exportins, as suggested by the reviewers, would considerably improve the study.

REVIEWER REPORTS

Reviewer #1:

Review of:
MSB-17-7608
Landscape of nuclear transport receptor cargo selectivity
Mackmull et al.

Summary:

The authors use BioID to identify proximity interactors of 16 nuclear transport receptors (NTR) in a

human cell line.

Review

This is a high quality mass spectrometry-based study conducted by the Beck laboratory. Using BioID to identify proximity interactions for 16 different nuclear transport receptors, the authors report >1000 new putative human NTR-cargo interactions.

I appreciate the fact that the authors used both N- and C-terminal BirA tagging, characterized the intracellular locations of the biotinylated protein partners for each of the baits used in the study, and utilized several different types of relevant BioID controls, to generate a high-confidence set of NTR interactors. The follow-up data analysis is also impressive, conducting comparisons to previously published datasets, and using a number of different analytical tools to evaluate the dataset in several different ways.

Major Concerns

1. I do, however, feel that there is something very important missing in this work - validation of novel biological insight.

A truly transformative manuscript of this type will highlight new biological insights provided by the dataset, then validate these insights, to show that the dataset is useful to the field, and to demonstrate how the dataset can be mined.

While I would not presume to direct the authors research, it could be very interesting here to see: (i) some examples of knock down/knock out of individual (or multiple) NTRs and the accompanying effects on transport of specific cargo proteins, based on the new predictions in this dataset. (ii) The authors suggest that their dataset reveals that only specific members of a given complex establish direct interactions with NTRs. They could validate this statement by demonstrating direct transporter-cargo binding in vitro using recombinant proteins, and/or using mutational analysis to disrupt interactions discovered in the dataset.

2. Do the data reveal any novel NTR binding sequences/structural motifs? Any biological surprises that could be further explored (quickly)?

3. While this is admittedly a significant amount of additional work, it would also dramatically increase the impact of the manuscript if the same analysis was conducted in an additional cell line.

Minor concerns:

1. While the terminology used here has been widely propagated in the literature, this reviewer suggests that it is incorrect to call the R118G BirA mutant protein "promiscuous". This word implies that the protein itself is biotinylating surrounding lysines, as opposed to the "non-promiscuous" WT protein, which recognizes a linear amino acid sequence in substrate polypeptides. It is not. The mutant enzyme is actually "abortive", prematurely releasing activated biotin (biotinoyl-AMP) which then diffuses away and chemically reacts with nearby amine groups.

2. Are the BirA-tagged NTRs properly localized? While the authors did a nice job of showing where the biotinylated proximity interactors are for each bait protein, I would also like to see either IF or live cell imaging for the endogenous versions of each NTR, along with the bait proteins themselves, in the same cells.

Reviewer #2:

Mackmull et al presents results of proteomic studies to identify protein cargos and other interacting partners of beta-Importin nuclear transport receptors (NTRs). Several proteomic studies of NTRs were reported in the last few years, but the current manuscript uses a different biochemical strategy of the BioID proximity ligation coupled to mass spectrometry. The Mackmull et al study is therefore a useful comparison with the previously published proteomic analyses. Their results also provide hints of direct interactors through identification of biotinylated peptides. Furthermore, the current

study avoids potential pitfalls of previous studies such as the use of digitonin-permeabilized cells and depletion of NTRs.

The authors analyzed proteomic results to provide useful and interesting new information and to provide support for surprising findings from previously reported studies. They show that $> 1/3$ of the proteome is involved in active nucleocytoplasmic transport. Such large fraction supports previous suggestions that a large fraction of the proteome (Kirli et al 2015) enters the nucleus at some time during the life of a cell.

The authors show functional redundancy of NTRs, especially amongst Importin-alphas and between homologous beta-Importins such as the IPO4/IPO5, TNPO1/TNPO2 pairs and between Importin-alphas and KPNB1. This is nice to see, but not unexpected. Perhaps the authors could provide more information with analysis of redundancy between different beta-Importins and also between the Exportins? The question of redundancy between very different NTRs is one that is often posed and the lack of knowledge causes confusion.

Finally, there is significant interest from biologists of various disciplines - eg. those studying signal transduction, cancer biologists, those studying nuclear processes, those studying biochemical interactions of NTRs and others, who will find lists of proteins that the authors find interacting with the different NTRs, useful. It may be useful for the authors to publish Tables of cargos (top scoring ones) for individual NTRs in the main part of the manuscript rather than have them be obscured in Supplementary Information.

1st Revision - authors' response

18 August 2017

We want to thank the reviewers for their very constructive criticism and suggestions that were very helpful to revise and improve our manuscript. Our detailed point by point response follows below.

Reviewer #1:

Review of: MSB177608: Landscape of nuclear transport receptor cargo selectivity. Mackmull et al.

Summary:

The authors use BioID to identify proximity interactors of 16 nuclear transport receptors (NTR) in a human cell line.

Review

This is a high quality mass spectrometry based study conducted by the Beck laboratory. Using BioID to identify proximity interactions for 16 different nuclear transport receptors, the authors report >1000 new putative human NTR cargo interactions. I appreciate the fact that the authors used both N and C terminal BirA tagging, characterized the intracellular locations of the biotinylated protein partners for each of the baits used in the study, and utilized several different types of relevant BioID controls, to generate a high confidence set of NTR interactors. The follow up data analysis is also impressive, conducting comparisons to previously published datasets, and using a number of different analytical tools to evaluate the dataset in several different ways.

Major Concerns

1. I do, however, feel that there is something very important missing in this work validation of novel biological insight. A truly transformative manuscript of this type will highlight new biological insights provided by the dataset, then validate these insights, to show that the dataset is useful to the field, and to demonstrate how the dataset can be mined.

While I would not presume to direct the authors research, it could be very interesting here to see: (i) some examples of knock down/knock out of individual (or multiple) NTRs and the accompanying effects on transport of specific cargo proteins, based on the new predictions in this dataset. (ii) The authors suggest that their dataset reveals that only specific members of a given complex establish direct interactions with NTRs. They could validate this statement by demonstrating direct transporter cargo binding in vitro using recombinant proteins, and/or using mutational analysis to disrupt interactions discovered in the dataset.

We agree with this critique and included the following additional data to demonstrate that our large scale analysis is a useful resource:

We have generated BirA fusion proteins for a selected subset of cargos that to the best of our knowledge have not been previously characterized. We reciprocally validated the respective transport pathways with a good success rate (described in lines 338 - 359 of the revised manuscript; Figures 5 and S4 and Tables S4, S5 and S8).*

As suggested by the reviewer, we quantified the nucleocytoplasmic distribution of two cargo proteins in response to gene silencing experiments for various NTRs (described in lines 359 - 366 of the revised manuscript; and Figure 5C, 5D, S4D, S4E, S4H). Also this data validates the specificity of the respective transport pathways.

In this context, we also looked on potential motifs (signal sequences) predicted for specific subunits of complexes (discussed in our response to the reviewer's point 2 right below).

2. Do the data reveal any novel NTR binding sequences/structural motifs? Any biological surprises that could be further explored (quickly)?

NLS and NES prediction generally suffers from low accuracy, possibly because these motifs are bipartite or at least to some extent act as 3D folds. Most likely because the available training data sets are rather small for the respective algorithms. Previous studies such as e.g. Kirli et al. have not done such analysis because it is considered challenging.

We have teamed up with Rob Russell from the University of Heidelberg and used his Dillimot algorithm to predict potential short linear motifs in significant enriched cargos of individual NTRs. As expected, this analysis recovered the PY-NLSs and cNLSs for transportin and importin alphas, respectively, to quite some extent, although Dillimot is not designed to identify bipartite motifs (Table S6). Mutational analysis of potential cNLSs yielded the expected cytoplasmic enrichment (Table S5, S6 and Figure 5E and 5F). Mass spectrometry data showed that the targeted subunits were integrated into the respective protein complexes (Table S4 and Figure 5A and S4B). We also identified DE-rich motifs in cargos of e.g. importin betas, which was unexpected (Table S6). We included some experimental validation of this motif but further analysis is required to understand if it is directly relevant for nuclear transport (Figure S4F and S4G). The results are summarized in Table S5 and described in lines 367 – 391 of the revised version of the manuscript.

3. While this is admittedly a significant amount of additional work, it would also dramatically increase the impact of the manuscript if the same analysis was conducted in an additional cell line.

Although we agree that such analysis would likely recover more cargos, we have to stress that this would be at least a year of work, including the molecular cloning and literally hundreds of MS runs. We thus hope that the reviewer agrees that such analysis is beyond the scope of the present manuscript.

Minor concerns:

1. While the terminology used here has been widely propagated in the literature, this reviewer suggests that it is incorrect to call the R118G BirA mutant protein "promiscuous". This word implies that the protein itself is biotinylating surrounding lysines, as opposed to the "nonpromiscuous" WT protein, which recognizes a linear amino acid sequence in substrate polypeptides. It is not. The mutant enzyme is actually "abortive", prematurely releasing activated biotin (biotinoylAMP) which then diffuses away and chemically reacts with nearby amine groups.

Agreed and corrected.

2. Are the BirA tagged NTRs properly localized? While the authors did a nice job of showing where the biotinylated proximity interactors are for each bait protein, I would also like to see either IF or live cell imaging for the endogenous versions of each NTR, along with the bait proteins themselves, in the same cells.

We have investigated this issue experimentally using the FLAG epitope that we had included in all our fusion proteins and compared the results to the Human Protein Atlas and literature. This analysis has been included into Figure 2B and indicates appropriate localization.

Reviewer #2:

Mackmull et al presents results of proteomic studies to identify protein cargos and other interacting partners of beta Importin nuclear transport receptors (NTRs). Several proteomic studies of NTRs were reported in the last few years, but the current manuscript uses a different biochemical strategy of the BioID proximity ligation coupled to mass spectrometry. The Mackmull et al study is therefore a useful comparison with the previously published proteomic analyses. Their results also provide hints of direct interactors through identification of biotinylated peptides. Furthermore, the current study avoids potential pitfalls of previous studies such as the use of digitonin permeabilized cells and depletion of NTRs.

The authors analyzed proteomic results to provide useful and interesting new information and to provide support for surprising findings from previously reported studies. They show that > 1/3 of the proteome is involved in active nucleocytoplasmic transport. Such large fraction supports previous suggestions that a large fraction of the proteome (Kirli et al 2015) enters the nucleus at some time during the life of a cell. The authors show functional redundancy of NTRs, especially amongst Importin alphas and between homologous beta Importins such as the IPO4/IPO5, TNPO1/TNPO2 pairs and between Importin alphas and KPNB1. This is nice to see, but not unexpected. Perhaps the authors could provide more information with analysis of redundancy between different beta Importins and also between the Exportins? The question of redundancy between very different NTRs is one that is often posed and the lack of knowledge causes confusion.

We have expanded the respective part of the main text (lines 290 to 304, see also Figure 4C and 4D) and included an additional display item that visualizes a global quantification of this aspect (Figure S3A). The overlap among exportins is rather low as compared to e.g. importin alphas. Careful inspection of our data revealed that there is an overlap of cargos between IPO4, IPO5 and XPO7.

Finally, there is significant interest from biologists of various disciplines eg. Those studying signal transduction, cancer biologists, those studying nuclear processes, those studying biochemical interactions of NTRs and others, who will find lists of proteins that the authors find interacting with the different NTRs, useful. It may be useful for the authors to publish Tables of cargos (top scoring ones) for individual NTRs in the main part of the manuscript rather than have them be obscured in Supplementary Information.

This was a good suggestion. We have included an additional display item into the revised version (Table 1). We also made an effort to improve the usability of the respective supplementary table containing large-scale data, e.g. by adding a color code (Table 4).

2nd Editorial Decision

5 October 2017

Thank you again for submitting your work to Molecular Systems Biology. We have now heard back from the two referees who accepted to evaluate the revised study. As you will see, the referees are now fully supportive and I am pleased to inform you that we will be able to accept your paper for publication in Molecular Systems Biology, pending the following minor modifications suggested by our editorial assistant at QC

REVIEWER REPORTS

Reviewer #1:

The authors have (more than) satisfactorily dealt with all of my previous issues / critiques.

Reviewer #2:

The reviewer is satisfied with the revisions.

YOU MUST COMPLETE ALL CELLS WITH A PINK BACKGROUND

Corresponding Author Name: Martin Beck
 Journal Submitted to: Molecular Systems Biology
 Manuscript Number: MSB-17-7608R